# Calculation and Analysis of Temperature Damage of Shimantan Concrete Gravity Dam Based on Macro–Meso Model

**DOI:** 10.3390/ma15207138

**Published:** 2022-10-13

**Authors:** Yantao Jiao, Liping Cheng, Ning Wang, Sizhe Wang, Luyao Ma

**Affiliations:** 1School of Water Conservancy, North China University of Water Resources and Electric Power, Zhengzhou 450045, China; 2College of Chemistry and Environmental Engineering, Pingdingshan University, Pingdingshan 467000, China; 3Nanjing Water Planning and Designing Institute Co., Ltd., Nanjing 210014, China

**Keywords:** concrete, random aggregate model, APDL commands flow, local coordinate system, mesomechanics

## Abstract

Considering that ANSYS software will automatically quit or the computer will freeze when generating random aggregate models of concrete by using some existing methods that are based on the ANSYS parametric design language (APDL), a new method of random aggregate placement using the ESEL command in APDL and the rotation of the local coordinate system is proposed in this paper. According to this method, a multiscale macroscopic and mesoscopic finite element model of the No. 9 non-overflow dam section of Shimantan dam is constructed. In addition, considering that most of the damage models adopted by the existing mesoscale simulation of concrete damage and fracture cannot take into account the interaction between aggregates, interfacial transition zone (ITZ), and mortar, an improved anisotropic temperature damage model is proposed in this paper. The aggregate placement simulation results show that the method presented in this paper can quickly generate two-dimensional (2D) random concrete aggregates, and the generation of three-dimensional (3D) aggregates can also be completed in a very short time, which can greatly improve the aggregate generation efficiency. Moreover, the aggregate shape generated by this method is very close to the real concrete aggregate shape. The crack propagation simulation results show that the sudden rise and fall of temperature can cause damage in the mortar and ITZ of concrete inside the dam body, which is the main reason for the generation of macroscopic through-cracks in the No. 9 non-overflow dam section of Shimantan dam during the operation period. Finally, it can be learned from the results that the method presented in this paper is reasonable and feasible, and can be extended to the crack propagation simulation of some other concrete gravity and arch dams.

## 1. Introduction

As one of the most important structural engineering materials, concrete has been widely used in various engineering fields. Among them, the cracking of mass concrete structures caused by temperature is also being paid more and more attention to by people in the fields of water conservancy and civil engineering. At present, the research on the working behavior and cracking of concrete structures under mechanical load is relatively thorough, and the research on the cracking of concrete structures at the macroscopic scale caused by temperature has also made many achievements, see, e.g., [1,2,3,4,5,6,7,8]. However, concrete is a multiphase composite material consisting of cement, sand, stone, and water. Thus, it may have some limitations in analyzing the deformation and cracking failure processes of concrete under load by ignoring its heterogeneity and assuming it to be a homogeneous material at the macroscopic level.

The mesoscopic mechanics method regards concrete as a three-phase heterogeneous composite material composed of aggregate, mortar matrix, and ITZ between the aggregate and the mortar matrix. By adopting an appropriate concrete mesoscopic structural model and dividing the finite element meshes at the mesoscopic level, the crack propagation and failure pattern of the concrete specimen can be simulated intuitively. Thus, the mesoscopic mechanics method is regarded as an effective way to solve the problem of concrete failure. It has been the subject of unprecedented attention and concern by many researchers and has achieved fruitful results. Buttignol et al. [9] established a concrete mesoscopic model by the Monte Carlo method to study the instantaneous creep problem of the compressive concrete specimen under high temperature conditions. The results show that the numerical simulation results are in good agreement with the test results, and the boundary conditions have an important effect on the total deformation of the concrete specimen. Wu et al. [10] employed the take-and-place approach to establish a reliable mesostructure model of concrete. In their work, based on the mesoscale model of concrete, not only the mechanical problems but also the effects of the interface debonding on the thermal conduction as well as humidity diffusion were investigated. By using the Monte Carlo method, Zhou et al. [11] established a random polygon mesoscale model of concrete, within which the mortar, ITZ, and aggregates are distinctly represented. The results show that the permeability of the ITZ and the aggregate volume fraction play a significant role in the freezing and thawing behavior of concrete. Chen et al. [12] proposed a heuristic packing algorithm for constructing 3D random aggregate models with high aggregate content. The results show that the algorithm is more efficient and robust. By using an in-house code based on Voronoi tessellation and splining techniques, Naderi et al. [13] developed a 3D mesoscale model of concrete composed of real-shape coarse aggregates, mortar, and ITZ. The results show that the irregular shape of the aggregate plays an important role in the micro-crack nucleation and ultimate fracture pattern. Rodrigues et al. [14] proposed a new approach for concurrent multiscale modeling of 3D crack propagation in concrete. By using this method, a macroscopic model with homogenized elastic parameters was adopted in the regions where the material behaves elastically. For regions where cracks were expected to occur, a mesoscopic model based on a mesh fragmentation technique was used to represent the concrete as a heterogeneous three-phase material composed of mortar matrix, aggregate, and ITZ. The simulation results show that the method can not only greatly reduce the calculation scale of the model, but also the propagation path of the crack is more consistent with the crack propagation path of the real concrete specimen. Thilakarathna et al. [15] summarized and commented on the evolution and research status of the concrete random aggregate model. They suggested scanning concrete specimens with laser scanning or computed tomography (XCT) technology to establish a real concrete particle structure and then performing structural calculations and analysis, which is a hot topic in current research. However, because of the complexity of generating a concrete random aggregate model and the high computational cost, only small-sized concrete specimens can be modeled, and large concrete structures cannot be modeled. In addition, many other researchers, such as the literature [16,17,18,19,20,21,22,23,24,25,26,27,28,29,30], have undertaken extensive studies on concrete mesoscopic mechanical models and developed various aggregate generation methods and placement algorithms.

Although the aggregate generation method and placement algorithm proposed by these studies can make the aggregate shape very close to the real concrete aggregate form and the aggregate filling amount can also reach a high degree, most of these studies need to program the aggregate generation and placement algorithm in Matlab, Fortran, Python, or C language to accomplish the aggregate placement. To complete the model visualization and finite element computation, researchers should employ additional finite element software, which requires a high level of programming and the development of a more complicated interface program to dock with this finite element software. Docking between programs usually requires transferring a large number of parameters, which occupies more computer memory. As a result, the demands on computer performance are considerable. Moreover, most of these studies are focused on cracking simulation of small-sized concrete specimens due to cost and computation time constraints. However, relevant reports for the simulation of crack propagation of 3D mass concrete structures with a large number of finite element meshes are relatively rare.

The finite element software ANSYS provides powerful pre-and post-processing capabilities. It also has an APDL programming language, similar to Fortran 77, with over 1000 commands. The APDL language can not only do most of the operations of the Fortran 77 language but can also perform other operations such as element selection, node selection, local coordinate system building, and rotation, which are not needed for secondary development by using other programming languages [31]. Therefore, researchers only need to master the APDL language to complete the programming implementation of aggregate generation and placement algorithms, as well as model visualization and finite element calculations, without having to learn additional languages or develop complex interface programs. Because of this advantage of the APDL language, many researchers have turned to this to do research on mesoscopic concrete models (e.g., [32,33,34]). Using the APDL language of ANSYS software, there are currently two ways to create a concrete mesoscopic mechanical model: first, the aggregate and ITZ boundaries are generated using Boolean operations, which are separated from the mortar boundary, and then the aggregate, ITZ, and mortar are meshed separately; second, the “background grid” method is used to generate aggregate and ITZ boundaries. That is, the mortar matrix is divided into finite element meshes first, and then the material properties of each element are judged by calculating the distance between all nodes on the element from the center of the aggregate sphere, and the corresponding material number is assigned to the element, so as to complete the aggregate placement. Although using these two methods to create a mesoscopic model of concrete has many of the benefits described above, it is worth noting that when using the first method (i.e., Boolean operations) to generate aggregates, mesh distortion is generally not caused if the meshing is carried out for large particle size particles, but when the aggregate size is small, it will often cause mesh distortion, making the calculation non-convergent and the program to report errors; when the second method is used, it is necessary to create a 3D array to store the node coordinate information of the element, and then judge the material properties of the element by calculating the distance from the node to the center of the aggregate sphere. This method can generate an aggregate model that meets the requirements if the number of elements is small, but if the number of elements reaches millions, the storage operation of the 3D array will occupy a large amount of computer memory space, reducing computing efficiency and increasing computer time, which is a significant challenge to the performance of the computer. Obviously, the random aggregate model of concrete established by the above two methods is also difficult to promote in practical engineering applications, especially in the damage-fracture analysis of high concrete dams. There are currently no relevant literature reports available.

From the above analysis, it can be seen that the existing research on mesoscopic mechanics method of concrete is usually limited to the simulation calculation of small-scale concrete specimens, while the cracking simulation of large-scale concrete structures, such as concrete gravity dams, is much less. To address the aforementioned challenges, this work presents a new approach for completing random aggregate placement using the ESEL command in the APDL language and adjusting the rotation angle of the local coordinate system.

When the method proposed in this paper is used, the placement boundary of aggregate is first determined, and then the radius information and the center coordinate information of aggregate are randomly generated by using the random number generation function RAND in the APDL language, and stored in a 2D array. Then, the mesh of the model is divided. After the mesh division is completed, the aggregate center coordinates and their radius information stored in the array are read, and the local coordinate system definition command LOCAL in the APDL language is used to define the local cartesian coordinate system, spherical coordinate system, or ellipsoidal coordinate system at any aggregate center coordinates. After the local coordinate system has been built, the ESEL command in the APDL language is used to screen the element set that meets the requirements of aggregate radius, and the material attribute of aggregate is given to this element set, thus completing the placement of an aggregate. By using the loop statement, traverse all aggregates stored in the array and read the corresponding aggregate center coordinates and radius data. According to the method mentioned above, all aggregates can be generated. As the APDL language of ANSYS software is used to complete the aggregate placement by using the method in this paper, and the ANSYS software platform is also used to complete the subsequent finite element calculation, the researchers can complete the aggregate generation, placement, model visualization, and finite element calculation without using other languages and compiling complex interface programs. Moreover, because there is no need to transfer a large number of parameters between different programs, the occupation of computer memory space can be greatly reduced. In addition, unlike some existing methods described above, when using the method in this paper, since the material properties of each element are not judged by calculating the distance from the node of each element to the center of the aggregate sphere, there is no need to define an oversized 3D array to store the node coordinate information of the element. As mentioned above, when the number of element nodes is large, the storage operation of the 3D array will occupy a large memory space on the computer, which is a great challenge to the performance of the computer, and this difficulty can be overcome by adopting the method proposed in this paper. To sum up, compared with the previous concrete aggregate placement methods, the use of this method in the process of concrete aggregate placement and subsequent finite element calculation can greatly reduce the occupation of computer memory space, improve the calculation efficiency, and make it possible to apply the mesoscopic mechanics method to the finite element calculation of large-scale 3D concrete structures.

In terms of material models, the material damage model or fracture model of each phase (e.g., [9,10,13,29]) that is used in the majority of current research on damage and fracture simulation of mesoscopic models of concrete is based on a macroscopic phenomenological method. That is, rather than the physical background of damage and fracture, or the mesoscopic structural changes within the material, it focuses on the effect of damage and fracture on the macroscopic mechanical properties of the material. Since the macroscopic phenomenological method simulates the macroscopic mechanical behavior of materials from macroscopic phenomena, the determination of the equations and parameters of these material models is usually half empirical and half theoretical, and has clear physical significance, which can directly reflect the stress state of structures. However, as described in the literature [35], the damage and failure of concrete are caused by the growth and accumulation of micro-cracks and micro-pores of various scales within it. Especially when considering temperature factors, due to the difference in the thermal expansion coefficient between aggregate and mortar, thermal incompatibility stress will be generated at the ITZ between mortar and aggregate, which will result in micro-cracks at the ITZ. When micro-cracks at the ITZ accumulate to a certain degree, it may extend to mortar and aggregate. Therefore, from the perspective of meso-mechanics, the micro-cracks, micro-defects, and other damage of concrete materials are interwoven and influenced by each other. When using the meso-mechanics method to investigate concrete damage and fracture problems, there are some limitations in rationality if the damage or fracture model used as material constitutive model is still derived using the traditional macroscopic phenomenological method, especially when considering temperature load. Therefore, an isotropic temperature damage model that takes into account the relationship between various phases of concrete has been proposed in the literature [35]. Considering the characteristics of anisotropic damage of concrete, this paper refers to relevant references and makes some improvements to this isotropic temperature damage model through a series of derivations, so an improved anisotropic temperature damage model is presented.

Based on the random aggregate placement method of concrete proposed in this paper, a 3D macro–meso finite element model of Shimantan concrete gravity dam is established in this paper, and the damage of the dam under two conditions of sudden temperature drop and sudden temperature rise is numerically simulated with the anisotropic temperature damage model presented in this paper, which takes into account the relationship between various phase materials of concrete. In order to verify the effectiveness of the aggregate placement method proposed in this paper, in Section 3, some small-scale concrete specimens are used to verify the placement efficiency and aggregate morphology. In addition, in order to verify the effectiveness of the anisotropic temperature damage model, the macro model and the bilinear damage model obtained based on the macro phenomenological method are also used to numerically simulate the temperature damage of Shimantan dam under the two working conditions. The results show that the method in this paper is feasible and can provide some references for other researchers to study similar projects.

## 2. Establishment of Concrete Random Aggregate Model

### 2.1. Application of Monte Carlo Method

The shape, size, and position of aggregates are all random in the random aggregate model. To describe this randomness, the Monte Carlo method is used to produce a group of uniformly distributed random variables *rdm* in the interval [0, 1], and random variables *RDM* in any other interval [a, b] can be calculated by using the following equation:(1)RDM=a+(b−a)×rdm

According to the Equation (1), the random position coordinates of the aggregate particles $\left(x_{i},y_{i},z_{i}\right)$ as well as the radius of the aggregate $R_{i}$ can be calculated. For ANSYS software, the RAND function in the APDL language can generate any random variable in the interval [a, b] without complex variations, making random number generation easier.

### 2.2. Fuller Aggregate Gradation Curve

Aggregate particle size distribution is an important parameter of the concrete random aggregate model, and the configuration of concrete according to the Fuller curve allows the concrete to have optimal compactness and macroscopic strength [32]. The expression for the Fuller curve is as follows:(2)$$P(d)=100\sqrt{d/d_{\max}}$$
where P(d) is the percentage of aggregate particles that pass through the sieve mesh size d; d is the mesh size of aggregate particles; and $d_{\max}$ is the maximum aggregate diameter.

### 2.3. Walraven Formula

The fuller curve, expressed by Equation (2) is mainly used to configure the aggregate gradation for the 3D problem. However, because the number of element meshes in the 3D concrete mesoscopic model increases significantly when compared to the 2D problem, the amount of computation is huge, requiring a lot of computer memory. Furthermore, researching the internal structure of concrete can provide people with intuitive knowledge and insight. To spatially simplify the 3D concrete mesoscopic model to a 2D problem, the author employs the calculation formula established by Walraven J. C. to convert Fuller’s 3D aggregate gradation curve into a 2D plane gradation. The expression is as follows:(3)$$P_{c}\left(d<d_{0}\right)=P_{k}(1.065d_{0}^{0.5}/d_{\max}^{0.5}-0.053d_{0}^{4}/d_{\max}^{4}-0.012d_{0}^{6}/d_{\max}^{6}-0.0045d_{0}^{8}/d_{\max}^{8}+0.0025d_{0}^{10}/d_{\max}^{10})$$
where $P_{k}$ is the percentage of aggregate volume in total volume; $d_{0}$ is the target particle sizes; $d_{\max}$ is the maximum aggregate diameter. By using Equation (3), the number of aggregates of each particle size on the cross-section of a 2D concrete specimen can be calculated.

### 2.4. Determination of the Centroid Coordinates and Radius of Spherical Aggregate Particles

To ensure that the aggregate particles do not exceed the boundary of the specimen, the centroid coordinate of each spherical aggregate particle is defined as follows:(4)$$\left\{\begin{array}{l} x_{\min}+R_{i}\leq x_{i}\leq x_{\max}-R_{i}\\ y_{\min}+R_{i}\leq y_{i}\leq y_{\max}-R_{i}\\ z_{\min}+R_{i}\leq z_{i}\leq z_{\max}-R_{i} \end{array}\right.$$
where $R_{i}$ represents the radius of the ith aggregate particle, of which the value range is determined by the aggregate gradation curve, and the value size is randomly generated by the RAND function; $x_{\min}$, $x_{\max}$, $y_{\min}$, $y_{\max}$, $z_{\min}$ and $z_{\max}$ are the boundary coordinate values of the specimen; $x_{i}$, $y_{i}$ and $z_{i}$ are the centroid coordinate components of the ith spherical aggregate particle, which is randomly generated by the RAND function under the conditions defined in Equation (4).

In order to prevent overlapping or intersecting between aggregates, a judgment condition should be introduced to eliminate unreasonable center coordinates of the spherical aggregate particles through looping statements. This determining condition can be expressed as follows:(5)$$\sqrt{{\left(x_{i}-x_{n}\right)}^{2}+{\left(y_{i}-y_{n}\right)}^{2}+{\left(y_{i}-y_{n}\right)}^{2}}\geq R_{i}+R_{n}$$
where n=1,2,⋯,i−1 represents the first to $\left(i-1\right)\mathrm{th}$ aggregates generated before the ith aggregate. Using Equation (5), the intersection of aggregates can be judged. Any aggregate of number i must not overlap the ones generated previously by number n. When the ith aggregate has not met the requirements of Equation (5), the program will jump out of the looping statements and regenerate the center coordinates of the ith aggregate by the RAND function under the conditions defined in Equation (4) until it meets the requirements of Equation (5).

## 3. Random Aggregate Generation Based on ANSYS

### 3.1. 2D Random Aggregate Model

When using the method presented in this paper to generate a 2D random aggregate model, the number of aggregates should be first determined according to Equation (3), and then the center coordinates and radius of any circular aggregate are randomly generated by using the RAND function of the APDL language, and saved in a 2D array agv (num, i). In the array agv (num, i), num represents the number of aggregate particles, and i (i = 1, 2, 3,) is used to store the center coordinates (x,y) and radius R, respectively. The center coordinates and radius of all aggregate particles are generated by looping statements, and in this process, whether the aggregate exceeds the boundary and whether there is a possibility of intersection or overlap of aggregate particles are judged by Equations (4) and (5). If the values of center coordinate and radius do not meet the conditions, they will be eliminated, and the appropriate values will be finally stored in the array agv (num, i).

When the values of center coordinate and radius of all aggregate particles are generated, the geometric model of aggregate particles is not generated first by using the method proposed in this paper. Otherwise, when the geometric model of mortar has been generated, it is necessary to use the Boolean operation to generate the boundary between the aggregate particles and mortar, which is contrary to the method in this article. Similar to the “background grid” method described earlier, when using the method in this paper, the geometric model of mortar is first generated and then the mesh is divided. Instead of defining a 3D array to store the coordinate information of all element nodes, the ESEL command in the APDL language and combining the rotation of the local coordinate system are used to select the corresponding aggregate and ITZ elements.

#### 3.1.1. Polygonal Aggregates

For polygonal aggregate, an element set of square aggregate, which takes the center of the circular aggregate as the centroid and the diameter of the circular aggregate as the length of the sides, is first selected by the command flow in the APDL language. The specific command flow is:esel,s,cent,x,agv(i,1) − agv(i,3), agv(i,1)+agv(i,3),0.1esel,r,cent,y,agv(i,2) − agv(i,3), agv(i,2)+agv(i,3),0.1

When a single square aggregate element set has been selected, by defining a local coordinate system at the center of the circular aggregate, the *xy* plane of the local coordinate system can rotate around the *z*-axis by a certain angle, which is randomly generated in the interval of (0, 90) using the RAND function. Then, in the newly generated square aggregate element set, the elements perpendicular to the x-axis of the new coordinate system and whose coordinate range is (*−R*, *R*) are selected through the command flow. In this way, parts of the elements are removed to form an aggregate element set similar to a polygon, as shown in Figure 1. The command flow used is as follows:local,11,0,agv(i,1),agv(i,2),0,rand(0,90)dsys,11esel,r,cent,x,-agv(i,3),agv(i,3),0.1mpchg,2,allallsel,allcsdele,all

Using a looping statement to traverse all aggregate particles and finally generate all polygonal aggregate particles. In this paper, for the selection of the ITZ elements, the adjacent nodes of all aggregate elements are first selected, and then the elements related to these nodes are selected, and their material parameter number is changed to ITZ material. The specific command flow is as follows:nsel,s,extnplot,0esln,s,1eplotmpchg,3,all

#### 3.1.2. Circular Aggregates

The process of choosing the elements that make up a circular aggregate is fairly straightforward. In this article, the local cylindrical coordinate system is first established at the center of the circular aggregate. Then, after choosing the nodes within the circle of radius *R*, the elements associated with the nodes are selected, and the material properties of these elements are modified to aggregate. The specific command flow is as follows:wpave,agv(i,1),agv(i,2)cswpla,11,1,1,1nsel,s,loc,x,0,agv(i,3),0.1esln,s,1mpchg,2,allallsel,allcsdele,all

For the selection of the ITZ elements, it is the same as above, and will not be repeated here.

#### 3.1.3. Elliptical Aggregates

For the selection of elliptical aggregate elements, the local cylindrical coordinate system is defined first at the center of the circle aggregate, and then the *xy* plane of the local coordinate system can be rotated around the *z*-axis by a certain angle, which is randomly generated in the interval of (0, 90) using the RAND function. After the above operations, the direction of the ellipse can be determined. During this process, the flatness of the ellipse can also be determined, i.e., the ratio of the radius length in the y-axis direction to the radius length in the x-axis direction. In this paper, this value is set to 0.6. Next, select the nodes in the ellipse with a radius of *R* on the *x*-axis, then select the elements associated with the nodes, and modify the material properties of these elements as aggregates. The specific command flow is:local,11,1,agv(i,1),agv(i,2),0,rand(0,180),,,0.6dsys,11nsel,s,loc,x,0,agv(i,3),0.1esln,s,1mpchg,2,allallsel,allcsdele,all

#### 3.1.4. Verification of Aggregate Placement Effect

In this calculation, 2D plane concrete specimens of two-graded concrete are selected for simulation. The side length of the specimen is 150 mm, the maximum aggregate particle size is 40 mm, the minimum particle size is 5 mm, and the volume fraction is taken as 0.65. The particle size is divided into two levels, 5–20 mm, and 20–40 mm. According to the Walraven’s formula expressed in Equation (3), the calculated distribution probability of aggregate for each particle size is shown in Table 1.

The area of the concrete specimen is: $A=150\times 150=22,500mm^{2}$. Assuming that the average particle size of level (20–40 mm) is 30 mm and the average particle size of level (5–20 mm) is 12.5 mm, the ratio of specimen area to aggregate area is shown in Table 2.

According to the data shown in Table 2, the number of level (20–40 mm) can be calculated as follows:$$n=\left(0.645-0.487\right)\times 31.85=5.0323\approx 6$$

In addition, the number of level (5–20 mm) can be calculated as follows:$$n=\left(0.487-0.230\right)\times 183.43=47.14\approx 48$$

After the aggregate number is determined, the polygonal aggregate model obtained by this method is shown in Figure 2a, the circular aggregate model is shown in Figure 2b, and the elliptical aggregate model is shown in Figure 2c.

### 3.2. 3D Random Aggregate Model

The generation of the 3D random aggregate model is similar to that of the 2D model. The difference is that when generating the 3D polyhedral aggregate, the local coordinate system needs to be rotated twice, i.e., the aggregate element set is first selected in the *xy* plane according to the method of the 2D model. Restore the local coordinate system to its default state. After that, rotate the local coordinate system 90 around the *y*-axis, and then rotate the local coordinate system at any angle around the *x*-axis, so that the previously selected aggregate element set can be further screened in the *xz* plane to form the element set conforming to the characteristics of polyhedral aggregate. For the generation of the spherical aggregate, only the local cylindrical coordinate system built in the 2D model at the spherical center needs to be changed into the local spherical coordinate system. For the generation of the ellipsoidal aggregate, it is necessary to first establish the local spherical coordinate system at the center of the aggregate, then respectively rotate the *xy* plane, *xz* plane, and *yz* plane of the local coordinate system at a random angle around the *z*-axis, *y*-axis, and *x*-axis to determine the direction of the ellipsoid, and finally determine the flatness of the ellipsoid, which is slightly different from the ellipsoidal aggregates in 2D. Here, not only is the ratio of the radius length of the *y*-axis to the radius length of the *x*-axis to be determined, but also the ratio of the radius length of the *z*-axis to the radius length of the *x*-axis is necessary to be determined. In this paper, both of the radios are taken as 0.6. The specific command flow is:local,11,2,agv(i,1),agv(i,2),agv(i,3),rand(0,180),rand(0,180),rand(0,180),0.6,0.6

The command flows such as selection of polyhedral or spherical aggregate elements are the same as for ellipsoidal aggregates and are not listed here.

It is assumed that the material is three-graded C20 concrete, and the concrete specimen size is 150 × 150 × 150 mm. For three-graded concrete, the particle size is divided into three levels, 40–80 mm, 20–40 mm, and 5–20 mm. According to Equation (2), each level of aggregate particle size can be calculated as follows: the number of level (40–80 mm) is 5, the number of level (20–40 mm) is 26, and the number of level (5–20 mm) is 425. Then the 3D aggregate placement is carried out by the method in this paper, and the generated 3D polyhedral, spherical, and ellipsoidal aggregate models are shown in Figure 3, Figure 4 and Figure 5, respectively.

It can be seen from Figure 2, Figure 3, Figure 4 and Figure 5 that the concrete random aggregate model generated by the method in this paper basically conforms to the geometric characteristics of concrete convex polyhedral aggregate, spherical aggregate, and spherical aggregate. When the aggregate content and computer configuration are the same during the modeling of a 3D random aggregate model, once the Boolean operation method is used to generate the aggregate geometric model, ANSYS software frequently prompts the mesh distortion during the subsequent mesh splitting process, and the program automatically exits. Another method is used to divide the element mesh first, then save the node information by defining 3D arrays and judging the material properties of the element. When the number of elements is small, this method can filter out aggregates and ITZ elements in a short time to complete the aggregate placement. However, when the number of elements reaches millions, the number of nodes will also reach millions. At this time, using the 3D array to store the coordinate information of the nodes will greatly occupy the memory space of the computer, making ANSYS software in a state of non-reaction for a long time, the computer freezes, which puts forward high requirements for the performance of the computer. However, when using the method proposed in this paper, it only takes 2 s to generate 2D aggregates and 5 min to generate 3D aggregates, which can greatly improve the efficiency of modeling. The reason is that, as mentioned in the introduction above, when the aggregate placement method proposed in this paper is used, it is unnecessary to transfer a large number of parameters between different programs, which can greatly reduce the occupation of computer memory space. In addition, when using the method in this paper, it is not necessary to define a large 3D array to store the node coordinate information of the element, which can also greatly reduce the occupation of computer memory space and improve the efficiency of aggregate placement.

## 4. Material Constitutive Model

Based on the strain equivalence hypothesis [36], the damage constitutive model under uniaxial conditions can be expressed as follows:(6)$$\sigma =(1-D)\sigma^{eff}=(1-D)E_{0}\varepsilon$$
where σ, $\sigma^{eff}$, D, $E_{0}$ and ε denote the nominal stress, the effective stress, the damage variable, the initial Young’s modulus and strain, respectively.

According to Equation (6), many damage evolution equations, including isotropic and anisotropic models, have been proposed by scholars from various countries, such as references [37,38,39,40,41]. However, as stated in the introduction section, these damage models are obtained on the basis of assuming that concrete is a homogeneous material and using the traditional macroscopic phenomenological method without considering the mutual influence of the damage between each phase of concrete. When using the mesoscopic mechanics method, which regards concrete as a multi-phase material consisting of aggregate, mortar, and ITZ, to simulate the damage and fracture of concrete, it is obviously inconsistent with the actual situation when using the damage model obtained from the macroscopic phenomenological method to simulate damage. Therefore, the literature [35] proposed a temperature damage model considering the relationship between each phase of concrete. The specific expression is:(7)$$\left\{\begin{array}{l} D_{m}=1-\frac{1}{1+\frac{0.00125G_{m}E_{m}}{f_{m}}{\left(1+A\Delta T\right)}^{2}}\\ D_{I}=1-\frac{1}{1+\frac{0.00125G_{I}E_{I}}{f_{I}}{\left(1+B\Delta T\right)}^{2}} \end{array}\right.$$
where $D_{m}$ and $D_{I}$ are the damage variables of mortar and ITZ in concrete, respectively. The value range of the damage variables is $0\leq D_{m},D_{I}\leq 1$, if $D_{m},D_{I}=0$, the material is not damaged, and if $D_{m},D_{I}=1$, it indicates that the material is completely fractured. $G_{m}$, $E_{m}$ and $f_{m}$ represent the fracture energy, elastic modulus, and tensile strength of the mortar, respectively. $G_{I}$, $E_{I}$ and $f_{I}$ represent the fracture energy, elastic modulus, and tensile strength of the ITZ, respectively. ΔT stands for the change of temperature. A and B can be expressed as follows:(8)$$\left\{\begin{array}{l} A=\frac{F_{1}\left(\alpha_{m}-\alpha_{a}\right)E_{a}\sqrt{E_{m}}}{C_{2}\left[E_{m}\left(1-\mu_{a}\right)+E_{a}\left(1+\mu_{m}\right)\right]}\\ B=\frac{\left(\alpha_{m}-\alpha_{a}\right)E_{a}\sqrt{E_{m}}}{C_{2}\left[E_{m}\left(1-\mu_{a}\right)+E_{a}\left(1+\mu_{m}\right)\right]} \end{array}\right.$$
where $F_{1}$ and $C_{2}$ are the fracture constants. Referring to the literature [42,43], in this paper, the values of the two constants are set to 1.414 and 0.2324, respectively. $\alpha_{m}$, $E_{m}$ and $\mu_{m}$ represent the thermal expansion coefficient, elastic modulus, and Poisson’s ratio of the mortar, respectively. $\alpha_{a}$, $E_{a}$ and $\mu_{a}$ denote the thermal expansion coefficient, elastic modulus, and Poisson’s ratio of the aggregate, respectively.

As stated in the literature [38,39], the damage of concrete material has obvious anisotropic characteristics. The damage evolution equation represented by Equation (7) reflects the damage condition of materials in one-dimensional (1D) space and is isotropic. In order to reflect the damage anisotropy of concrete material, Equation (7) needs to be extended to 3D space and be directional. When the anisotropic damage variable is used instead of the scalar damage variable, Equation (6) needs to be extended accordingly and the corresponding stiffness degradation matrix should be defined. After being extended, Equation (6) can be expressed as follows:(9)$$\sigma^{eff}={\left(I-\omega \right)}^{-1}\cdot \sigma$$
where $\sigma^{eff}$ is the effective stress tensor, σ is the nominal stress tensor, I is the second-order identity tensor, and ω is the second-order damage tensor, which can be expressed as follows:(10)$$\omega =\sum_{j=1}^{3}\omega_{i}({\vec{n}}_{j}⊗{\vec{n}}_{j})$$
where ${\vec{n}}_{j}$ is the main direction of damage vector and $\omega_{i}$ is the principal value of the damage tensor ω. In the principal damage space, the matrix representation of the tensor ω can be expressed as follows:(11)$$\left[\omega_{ij}\right]=\left[\begin{array}{ccc} {\hat{\omega }}_{1} & & \\ & {\hat{\omega }}_{2} & \\ & & {\hat{\omega }}_{3} \end{array}\right]$$
where ${\hat{\omega }}_{i},\;i=1,2,3$ represents the eigenvalues and can be expressed as:(12)$${\hat{\omega }}_{i}=1-\frac{E_{i}}{E_{0}},\;i=1,2,3$$
with $E_{i}$ being an unknown quantity to be determined by the strain-based damage evolution function. In the subsequent development, the superscript hat symbol $\left(\hat{•}\right)$ denotes a principal value of $\left(\hat{•}\right)$.

As described by Murakami [44], the effective stress tensor obtained by the strain equivalence hypothesis (i.e., Equation (9)) is asymmetric. Since it is usually inconvenient to use the asymmetric effective stress tensor in the formulation of constitutive and evolution equations, several symmetrized methods have been proposed by many authors. Among these methods, one can use the strain energy equivalence hypothesis to overcome this limitation. However, as stated in the literature [29], using the strain energy equivalence hypothesis will complicate the constitutive model. The multiplicative decomposition symmetrization method proposed by the literature [45] is used frequently, such as in the literature [39,46,47,48]. By this method, Equation (9) can be rewritten as follows:(13)$$\sigma^{eff}={\left(I-\omega \right)}^{-1/2}\sigma {\left(I-\omega \right)}^{-1/2}$$
or
(14)$$\sigma =M\left(\omega \right):\sigma^{eff}$$

$M\left(\omega \right)$ is a fourth-order damage-effect tensor and can be expressed as:(15)$$M_{ijkl}=\frac{1}{2}[{\left(\delta_{ik}-\omega_{ik}\right)}^{1/2}{\left(\delta_{jl}-\omega_{jl}\right)}^{1/2}+{\left(\delta_{il}-\omega_{il}\right)}^{1/2}{\left(\delta_{jk}-\omega_{jk}\right)}^{1/2}]$$

In the principal coordinate system of damage ω with the Voigt notations, the fourth-order damage-effect tensor M can be expressed as the following “diagonal matrix form”:(16)$$\left[M(\hat{\omega })\right]=dig[\phi_{1}\;\;\phi_{2}\;\;\phi_{3}\;\;\sqrt{\phi_{2}\phi_{3}}\;\;\sqrt{\phi_{1}\phi_{3}}\;\;\sqrt{\phi_{1}\phi_{2}}]$$
where $\phi_{i}^{}=1-{\hat{\omega }}_{i}^{}$ with ${\hat{\omega }}_{i}^{}(i=1,2,3)$ being the principal damage variable.

Since the fourth-order damage-effect tensor M contained more individual components, some authors [49,50] assumed that the principal axes of damage coincide with the principal axes of stress and strain and used the simplified damage effect tensor in Equation (16) to establish the relation between the effective and nominal stress tensors, and the matrix representation of the nominal stress tensor can be written as:(17)$$\left\{\sigma \right\}=\left[M\left(\hat{\omega }\right)\right]{\left\{\sigma \right\}}^{eff}=\left[M\left(\hat{\omega }\right)\right]{\left[E\right]}_{0}\left\{\varepsilon \right\}={\left[E\right]}_{d}\left\{\varepsilon \right\}$$
where ${\left[E\right]}_{0}$ is the initial undamaged elastic stiffness matrix and ${\left[E\right]}_{d}$ is the damaged elastic stiffness matrix which can be expressed as:(18)$${\left[E\right]}_{d}=\varpi \cdot \left[\begin{array}{cccccc} E_{1}\left(1-\mu \right) & E_{1}\mu & E_{1}\mu & 0 & 0 & 0\\ E_{2}\mu & E_{2}\left(1-\mu \right) & E_{2}\mu & 0 & 0 & 0\\ E_{3}\mu & E_{3}\mu & E_{3}\left(1-\mu \right) & 0 & 0 & 0\\ 0 & 0 & 0 & \frac{E_{23}\left(1-2\mu \right)}{2} & 0 & 0\\ 0 & 0 & 0 & 0 & \frac{E_{13}\left(1-2\mu \right)}{2} & 0\\ 0 & 0 & 0 & 0 & 0 & \frac{E_{12}\left(1-2\mu \right)}{2} \end{array}\right]$$
with $E_{i}=\phi_{i}E_{0}$, $E_{ij}=\sqrt{\phi_{i}\phi_{j}}E_{0}\left(i,j=1,2,3\right)$ and $\varpi =1/\left(1+\mu \right)\left(1-2\mu \right)$.

In addition, $\left\{\sigma \right\}$ and $\left\{\varepsilon \right\}$ in Equation (17) can be expressed as:(19)$$\begin{array}{l} \left\{\sigma \right\}={\left[\begin{array}{cccccc} \sigma_{11} & \sigma_{22} & \sigma_{33} & \sigma_{23} & \sigma_{13} & \sigma_{12} \end{array}\right]}^{T}\\ \left\{\varepsilon \right\}={\left[\begin{array}{cccccc} \varepsilon_{11} & \varepsilon_{22} & \varepsilon_{33} & \varepsilon_{23} & \varepsilon_{13} & \varepsilon_{12} \end{array}\right]}^{T} \end{array}$$

The damage–effect matrix shown in Equation (16) is in the principal coordinate system of damage, which corresponds to the direction of the principal strains. In terms of a general coordinate system, it should be transformed to the global coordinate system as follows:(20)$${\left[M\left(\hat{\omega }\right)\right]}_{S}={\left[T\right]}^{T}\left[M\left(\hat{\omega }\right)\right]\left[T\right]\;$$
where
(21)$$\left[T\right]=\left[\begin{array}{cccccc} l_{1}^{2} & m_{1}^{2} & n_{1}^{2} & l_{1}m_{1} & m_{1}n_{1} & n_{1}l_{1}\\ l_{2}^{2} & m_{2}^{2} & n_{2}^{2} & l_{2}m_{2} & m_{2}n_{2} & n_{2}l_{2}\\ l_{3}^{2} & m_{3}^{2} & n_{3}^{2} & l_{3}m_{3} & m_{3}n_{3} & n_{3}l_{3}\\ 2l_{1}l_{2} & 2m_{1}m_{2} & 2n_{1}n_{2} & l_{1}m_{2}+l_{2}m_{1} & m_{1}n_{2}+m_{2}n_{1} & n_{1}l_{2}+n_{2}l_{1}\\ 2l_{2}l_{3} & 2m_{2}m_{3} & 2n_{2}n_{3} & l_{2}m_{3}+l_{3}m_{2} & m_{2}n_{3}+m_{3}n_{2} & n_{2}l_{3}+n_{3}l_{2}\\ 2l_{3}l_{1} & 2m_{3}m_{1} & 2n_{3}n_{1} & l_{3}m_{1}+l_{1}m_{3} & m_{3}n_{1}+m_{1}n_{3} & n_{3}l_{1}+n_{1}l_{3} \end{array}\right]$$

By substituting Equation (20) into Equation (17), Equation (17) can be rewritten as:(22)$$\left\{\sigma \right\}={\left[M\left(\hat{\omega }\right)\right]}_{S}{\left\{\sigma \right\}}^{eff}={\left[M\left(\hat{\omega }\right)\right]}_{S}{\left[E\right]}_{0}\left\{\varepsilon \right\}$$

Then the nominal stress calculated by Equation (22) is the physical quantity in the global coordinate system. After the damaged elastic stiffness matrix has been defined, it is necessary to determine the damage evolution equation that can calculate the anisotropic damage variables ${\hat{\omega }}_{i}^{}(i=1,2,3)$ It can be seen from Equation (7) that the temperature damage evolution equation proposed in reference [35] is a function of temperature, which is a scalar variable and has no direction. Therefore, it cannot well express the anisotropic damage characteristics of concrete under the multiaxial stress state and should be extended to a multi-axial form. As mentioned in the literature [51], the surface deformation of the concrete dam blocks caused by temperature change will be constrained by the concrete inside the dam blocks, and the constraint coefficient can be expressed as:(23)$$\gamma =\frac{\varepsilon }{\alpha \Delta T}$$
where γ is the constraint coefficient, α is the thermal expansion coefficient of the concrete, and ΔT denotes the temperature increment.

According to Equation (23), one can get that:(24)$$\Delta T=\frac{\varepsilon }{\alpha \gamma }$$

By substituting Equation (24) into Equation (7), Equation (7) can be rewritten as follows:(25)$$\left\{\begin{array}{l} D_{m}=1-\frac{1}{1+\frac{0.00125G_{m}E_{m}}{f_{m}}{\left(1+A\frac{\varepsilon }{\alpha_{m}\gamma }\right)}^{2}}\\ D_{I}=1-\frac{1}{1+\frac{0.00125G_{I}E_{I}}{f_{I}}{\left(1+B\frac{\varepsilon }{\alpha_{I}\gamma }\right)}^{2}} \end{array}\right.$$
where $\alpha_{m}$ and $\alpha_{I}$ represent the thermal expansion coefficient of the mortar and ITZ, respectively.

Since the strain ε in Equation (25) is a scalar, the damage variables obtained from Equation (25) are still isotropic variables. Considering the characteristics of anisotropic damage, the principal strain $\varepsilon_{i}\left(i=1,2,3\right)$ can be used to replace the strain ε in the Equation (25) to make the damage evolution equation have direction. Then Equation (25) can be changed to:(26)$$\left\{\begin{array}{l} {\hat{\omega }}_{m\left(i\right)}=1-\frac{1}{1+\frac{0.00125G_{m}E_{m}}{f_{m}}{\left(1+A\frac{{\hat{\varepsilon }}_{i}}{\alpha_{m}\gamma }\right)}^{2}}\\ {\hat{\omega }}_{I\left(i\right)}=1-\frac{1}{1+\frac{0.00125G_{I}E_{I}}{f_{I}}{\left(1+B\frac{{\hat{\varepsilon }}_{i}}{\alpha_{I}\gamma }\right)}^{2}} \end{array}\right.\;$$
where i=1,2,3; ${\hat{\varepsilon }}_{i}$ is the principal strain; ${\hat{\omega }}_{m\left(i\right)}$ and ${\hat{\omega }}_{I\left(i\right)}$ are the damage values of mortar and ITZ in the three principal strain directions, respectively; γ is the constraint coefficient, which is set to 0.95 in this paper based on the literature [51].

Obviously, the damage evolution equation expressed in Equation (26) can reflect the damage anisotropy of concrete material, but when the principal strain is compressive strain, the calculated damage value may deviate from the actual situation. As stated in the literature [39,52], the damage of concrete material is mainly caused by tensile strain. When the material is under tensile loading, the micro-cracks propagate perpendicular to the direction of tension. The direction of damage is the same as the direction of tension and is called “direct damage.” The damage caused by material under compression is mainly caused by lateral tension strain due to the Poisson effect. The direction of damage is orthogonal to the direction of compression, and the direction of microcrack propagation is parallel to the direction of loading, which is called “indirect damage.” Therefore, further improvement is required for Equation (26).

When the finite element software is used for structural calculation, the user-defined constitutive model can be realized via the user material subroutine of the software. ABAQUS software provides user material subroutine UMAT, ANSYS software provides user material subroutine USERMAT, and some other finite element software also provides corresponding user material subroutines, which will not be listed one by one here. When users use these material subroutines to define their own material constitutive model, the Jacobian matrix of material constitutive, which is the change rate of stress increment corresponding to strain increment, needs to be provided. For linear elastic material constitutive models, such as the linear elastic damage model defined in this paper, the Jacobian matrix is ${\left[M\left(\hat{\omega }\right)\right]}_{S}{\left[E\right]}_{0}$ in Equation (22). For other elastic–plastic or more complex models, the corresponding Jacobian matrix can be defined according to the characteristics of the model. After the Jacobian matrix is defined, the main program usually uses the geometric relationship to calculate the strain increment Δε of each incremental step or iteration, which is transmitted to the user material subroutine. The user material subroutine calculates the stress increment of each incremental step or iteration according to the Jacobian matrix and obtains the updated stress and strain with the following equation:(27)$$\left\{\begin{array}{l} \sigma^{n+1}=\sigma^{n}+\Delta \sigma \\ \varepsilon^{n+1}=\varepsilon^{n}+\Delta \varepsilon \end{array}\right.$$
where n represents incremental step.

According to the above calculation principle, under triaxial stress states, the normal stress of any point in the structure along one coordinate axis will be affected by the strain in the other two coordinate axes that are orthogonal to it when the stress–strain calculation and analysis of the structure are carried out by the finite element method. The value of the normal stress along any coordinate axis is actually the result of the strain superposition along the three coordinate axes. Any principal stress at this point is also the result of the superposition of three principal strains. When damage is not considered, the principal stress at any point in the principal stress space can be calculated by the following formula:(28)$$\left\{\begin{array}{l} {\hat{\sigma }}_{1}\\ {\hat{\sigma }}_{2}\\ {\hat{\sigma }}_{3} \end{array}\right\}=\frac{E_{0}}{\left(1+\mu \right)\left(1-2\mu \right)}\left[\begin{array}{ccc} 1-\mu & \mu & \mu \\ \mu & 1-\mu & \mu \\ \mu & \mu & 1-\mu \end{array}\right]\left\{\begin{array}{c} {\hat{\varepsilon }}_{1}\\ {\hat{\varepsilon }}_{2}\\ {\hat{\varepsilon }}_{3} \end{array}\right\}$$

According to the general Hooke’s law, the strain components in the three principal directions are satisfied with the following relationships:(29)$$\begin{array}{l} {\hat{\varepsilon }}_{i}=\varepsilon_{ii}+\varepsilon_{ij}+\varepsilon_{ik}=\frac{1}{E_{0}}\left({\hat{\sigma }}_{i}-\mu {\hat{\sigma }}_{j}-\mu {\hat{\sigma }}_{k}\right)\;\;\\ \left(i,j,k=1,2,3\;\mathrm{and}\;i\neq j\neq k\right) \end{array}$$
where $\varepsilon_{ii}={\hat{\sigma }}_{i}/E_{0}$ is the strain in the *i*-direction caused by the stress in the *i*-direction; $\varepsilon_{ij}=-\mu {\hat{\sigma }}_{j}/E_{0}$ is the strain in the *i*-direction caused by the stress in the *j*-direction due to the Poisson effect; $\varepsilon_{ik}=-\mu {\hat{\sigma }}_{k}/E_{0}$ is the strain in the *i*-direction caused by the stress in the *k*-direction due to the Poisson effect.

According to Equation (29), Equation (28) can also be expressed as:(30)$$\left\{\begin{array}{l} {\hat{\sigma }}_{1}\\ {\hat{\sigma }}_{2}\\ {\hat{\sigma }}_{3} \end{array}\right\}=E_{0}\left\{\begin{array}{c} \varepsilon_{11}\\ \varepsilon_{22}\\ \varepsilon_{33} \end{array}\right\}$$

It can be seen from Equation (29) that when ${\hat{\sigma }}_{j}={\hat{\sigma }}_{k}=0$, one can get ${\hat{\varepsilon }}_{i}=\varepsilon_{ii}={\hat{\sigma }}_{i}/E_{0}$. This implies that the magnitude of a principal stress is only related to the principal strain under the uniaxial stress state, which is different from the triaxial stress state. Therefore, Equation (30) can actually be regarded as the expression of the stress–strain relationship of three principal stresses under the uniaxial stress state. Combining Equation (28), one can get that, under the triaxial stress state, when the three principal strains are all greater than 0 (i.e., ${\hat{\varepsilon }}_{i}>0\;\;\left(i=1,2,3\right)$), any principal stress ${\hat{\sigma }}_{i}$ reaches a certain value, such as the peak strength of the material, the required value of the strain ${\hat{\varepsilon }}_{i}$ may be smaller than the strain $\varepsilon_{ii}$ under the uniaxial stress state. This is because under the triaxial stress state, the principal stress ${\hat{\sigma }}_{i}$ is the result of the superposition of the principal strains in three directions, and its corresponding strain can be expressed as ${\hat{\varepsilon }}_{i}=\varepsilon_{ii}+\varepsilon_{ij}+\varepsilon_{ik}$. However, under the uniaxial state, the principal stress ${\hat{\sigma }}_{i}$ in one direction is only related to the strain $\varepsilon_{ii}$ in the corresponding direction. When the principal strain in one direction is tensile strain and the other two directions are compressive strain (i.e., ${\hat{\varepsilon }}_{i}>0\;\;\left(i=1,2,3\right)$), the required value of the strain ${\hat{\varepsilon }}_{i}$ for the principal stress ${\hat{\sigma }}_{1}$ to reach the peak strength may be larger than the strain $\varepsilon_{ii}$ under the uniaxial stress state. Equation (7) proposed by reference [35] is the uniaxial damage evolution equation derived from the test results under the uniaxial stress state. Obviously, using Equation (25) improved from Equation (7) to calculate the damage under the uniaxial stress state will be consistent with the test results, but under the multiaxial stress state, only by giving the directionality of strain in Equation (26) to calculate the damage value in all directions, it may be greatly different from the test results. For example, under the multiaxial stress state, when the three principal strains are greater than 0 (i.e., ${\hat{\varepsilon }}_{i}>0\;\;\left(i=1,2,3\right)$), and at the time that the strain in a certain direction reaches ${\hat{\varepsilon }}_{i}$, the stress ${\hat{\sigma }}_{i}$ in the corresponding direction calculated by the program reaches the peak strength of concrete, but under the uniaxial state, the stress calculated by this strain value ${\hat{\varepsilon }}_{i}$ may not reach the peak strength of concrete. At this time, the damage value ${\hat{\omega }}_{\left(i\right)}$ calculated by substituting the strain value ${\hat{\varepsilon }}_{i}$ into Equation (26) will be smaller than the actual damage value. This is because in the actual uniaxial test, when the stress reaches the peak strength, the strain corresponding to the real damage degree inside the material is the peak strain, while the aforementioned strain ${\hat{\varepsilon }}_{i}$, which is substituted into Equation (26) to calculate the damage value, has not reached the peak strain, the calculated damage value cannot reflect the true damage degree of the structure. In other words, according to the stress calculation results, the tensile strength of the material has been reached in one direction, which indicates that fracture failure may have occurred in the structure, but the damage calculation results indicate that the damage of the structure does not reach the extent of fracture failure.

It can be seen from Equation (29) that for the same stress value ${\hat{\sigma }}_{i}$, ${\hat{\varepsilon }}_{i}$ is the strain corresponding to the stress value ${\hat{\sigma }}_{i}$ under the triaxial stress state, whereas $\varepsilon_{ii}$ is the strain corresponding to the stress value ${\hat{\sigma }}_{i}$ under the uniaxial stress state. If substituting $\varepsilon_{ii}$ for ${\hat{\varepsilon }}_{i}$ in Equation (26), then the calculated principal damage results can match the uniaxial test results and truly reflect the damage degree of the structure. Based on the above analysis, $\varepsilon_{ii}$ can be defined as follows:

1.When all principal stresses are positive, i.e., tension–tension–tension mode, then $\varepsilon_{ii}$ can be expressed as follows:

(31)$$\varepsilon_{ii}={\hat{\varepsilon }}_{i}-\varepsilon_{ij}-\varepsilon_{ik}={\hat{\varepsilon }}_{i}+\frac{1}{E_{0}}\left(\mu {\hat{\sigma }}_{j}+\mu {\hat{\sigma }}_{k}\right)\;$$
where i,j,k=1,2,3 and i≠j≠k; ${\hat{\varepsilon }}_{i}$ is the principal strain in *i*-direction calculated by the user material subroutine; ${\hat{\sigma }}_{j}$ and ${\hat{\sigma }}_{k}$ are the principal stresses in *j*- and *k*-directions calculated by the user material subroutine, respectively.

2.When two principal stresses are positive and one principal stress is negative, i.e., tension–tension–compression mode, it can be seen from Equation (30) that the strain $\varepsilon_{ii}$ corresponding to the direction of compressive stress must be negative. As mentioned above, the damage in the compression direction is mainly caused by the lateral tensile strain due to the Poisson effect. The damage direction is orthogonal to the compression direction, and the microcrack propagation direction is parallel to the loading direction. Because the lateral deformation caused by the tensile stress is usually compressive deformation, which makes little contribution to the damage in the direction of compressive stress. Therefore, it is assumed that the damage in the direction of compression under the tension-tension-compression state is only related to the tension strain caused by the Poisson effect in the orthogonal direction but not to the strains in the other two directions. In addition, then $\varepsilon_{ii}$ can be expressed as follows:

(32)$$\varepsilon_{ii}=\{\begin{array}{l} \varepsilon_{ii}^{+}=\langle {\hat{\varepsilon }}_{i}+\frac{1}{E_{0}}\left(\mu {\hat{\sigma }}_{j}+\mu {\hat{\sigma }}_{k}\right)\rangle \;\;\;\left({\hat{\sigma }}_{i}>0,{\hat{\sigma }}_{j}>0,{\hat{\sigma }}_{k}<0\right)\\ \varepsilon_{ii}^{-}=\frac{-\mu {\hat{\sigma }}_{i}}{E_{0}}\;\;\;\;\;\;\;\;\;\;\;\;\;\;\;\;\;\;\;\left({\hat{\sigma }}_{i}<0,{\hat{\sigma }}_{j}>0,{\hat{\sigma }}_{k}>0\right) \end{array}$$
where i,j,k=1,2,3 and i≠j≠k; $\varepsilon_{ii}^{+}$ is the tensile damage strain corresponding to tensile stress; $\varepsilon_{ii}^{-}$ is the equivalent transfer tensile damage strain corresponding to compressive stress; the sign $\langle •\rangle$ indicates Macauley bracket, which is defined as $\langle x\rangle =\frac{1}{2}(\left|x\right|+x),k=1,2,3$.

3.When one principal stress is positive and the other two principal stresses are negative, i.e., tension–compression–compression mode, the damage in one compressive direction is mainly caused by the combined action of lateral deformation caused by the Poisson effect in the other two directions. In addition, in this case, the other compressive direction contributes to lateral tensile deformation while the direction of tensile stress contributes to lateral contraction deformation. In this state, the following equation is used to express $\varepsilon_{ii}$ in this paper:

(33)$$\varepsilon_{ii}=\left\{\begin{array}{l} \varepsilon_{ii}^{+}=\langle {\hat{\varepsilon }}_{i}+\frac{1}{E_{0}}\left(\mu {\hat{\sigma }}_{j}+\mu {\hat{\sigma }}_{k}\right)\rangle \;\;\;\left({\hat{\sigma }}_{i}>0,{\hat{\sigma }}_{j}<0,{\hat{\sigma }}_{k}<0\right)\\ \varepsilon_{ii}^{-}=\langle \frac{1}{E_{0}}\left(\mu \left|{\hat{\sigma }}_{k}\right|-\mu {\hat{\sigma }}_{j}\right)\rangle \;\;\;\;\;\;\left({\hat{\sigma }}_{i}<0,{\hat{\sigma }}_{j}>0,{\hat{\sigma }}_{k}<0\right) \end{array}\right.$$
where i,j,k=1,2,3 and i≠j≠k.

4.When all three principal stresses are negative, i.e., compression–compression–compression mode, the damage in one direction is mainly caused by the combined action of lateral deformation caused by the Poisson effect in the other two directions. The other two compressive directions contribute to lateral tensile deformation in this direction. Therefore, in this state, $\varepsilon_{ii}$ can be expressed as follows:

(34)$$\varepsilon_{ii}=\varepsilon_{ii}^{-}=\frac{1}{E_{0}}\left(\mu \left|{\hat{\sigma }}_{k}\right|+\mu \left|{\hat{\sigma }}_{j}\right|\right)\;\;\;\;\;\;\;\;\left({\hat{\sigma }}_{i}<0,{\hat{\sigma }}_{j}<0,{\hat{\sigma }}_{k}<0\right)$$
where i,j,k=1,2,3 and i≠j≠k.

By substituting $\varepsilon_{ii}$ defined by Equations (31)–(34) into Equation (26), one can get the following expression:(35)$$\left\{\begin{array}{l} {\hat{\omega }}_{m\left(i\right)}=1-\frac{1}{1+\frac{0.00125G_{m}E_{m}}{f_{m}}{\left(1+A\frac{\varepsilon_{ii}}{\alpha_{m}\gamma }\right)}^{2}}\\ {\hat{\omega }}_{I\left(i\right)}=1-\frac{1}{1+\frac{0.00125G_{I}E_{I}}{f_{I}}{\left(1+B\frac{\varepsilon_{ii}}{\alpha_{I}\gamma }\right)}^{2}} \end{array}\right.\;\;$$

The damage evolution equation represented by Equation (35) can reflect the damage anisotropy of concrete material. By substituting the calculation result of ${\hat{\omega }}_{\left(i\right)}$ into Equation (22), the elastic stiffness matrix considering damage can be calculated and the stress–strain calculation of the structure can be completed. Equation (22) and Equation (35) are the main equations of the anisotropic temperature damage model for concrete proposed in this paper. Finally, the proposed model is implemented in the ANSYS software via the user subroutine USERMAT.

## 5. Calculation and Analysis of Temperature Damage of Concrete Gravity Dam

### 5.1. Project Overview

The Shimantan Reservoir is located on the rolling river of the Honghe tributary upstream of the Huaihe River in Wugang City, Henan Province. The original reservoir collapsed due to the “75·8” catastrophic flood. Reservoir reconstruction works commenced in September 1993, and the main part was completed in December 1997. The Shimantan water conservancy hinge project is composed of the barrage, spillway, water intake structure, power station, and other structures. With a total reservoir capacity of 120 million m^3^, it is a large-scale water conservancy project mainly for industrial water supply and flood control, combined with comprehensive utilization of irrigation, tourism, and aquaculture. The barrage dam is a concrete gravity dam with a maximum dam height of 40.5 m, a crest elevation of 112.50 m and a crest length of 645 m. It is divided into 22 dam sections with a length of 16.0–42.0 m. Nos. 1–9 dam section is a 320.0 m right bank non-overflow dam section, Nos. 10-16 dam section is a 132.0 m overflow dam section, and Nos. 17–22 dam section is a 193.0 m left bank non-overflow dam section, with No. 19 dam section being a bottom hole dam section measuring 18.0 m [53]. As the dam site area is located in the temperate monsoon climate zone, no thermal insulation measures were set on the dam body surface in design. However, since the reconstruction of Shimantan Reservoir was put into operation, there have been many cracks in the dam body, especially several through cracks distributed vertically on the upper part of the dam body with good regularity. On the upstream surface (water level above 108.20 m), 39 vertical cracks along the flow direction on the top surface of the dam and 77 vertical cracks on the downstream surface of the dam, and most of the cracks on the dam top correspond to the position of the vertical cracks on the upstream and downstream sides. Relevant data indicate that most of these dam body cracks occur during the operation period after impounding. There are two penetrating cracks running through the upstream, dam crest and downstream of No. 9 non-overflow dam section of the dam. In this paper, the No. 9 non-overflow dam section is selected as a typical dam section, the temperature field and damage field during the operation period are calculated, and the cause of dam body cracks is analyzed. The section sketch and material partition of No. 9 non-overflow dam section are shown in Figure 6.

### 5.2. 3D Finite Element Model

In this paper, the thermal structural coupling analysis of No. 9 non-overflow dam section is carried out by using ANSYS software. For the 3D finite element model, the specified calculation coordinate system is: the *x*-axis is the flow direction along the river (vertical to the dam axis), and the direction pointing downstream is positive; the *y*-axis is vertical, pointing upward is positive; the *z*-axis is parallel to the dam axis, and it is positive when pointing to the right bank. The scope of the calculation model determined is: in the *x*-direction, taking the dam axis as the zero point, and the upstream and downstream cut-off boundaries are all taken as 1.5 times the dam height (the dam height is 39.5 m); the height of the dam foundation is taken as 1.5 times the dam height in the *y*-direction; and in the *z*-direction, taking a dam section with a length of 42 m. In the thermal structure coupling analysis, the thermal element type is solid70, and the structural element is solid45. The two types of elements are used together. After the temperature field analysis is completed, solid70 is converted to solid45 to calculate the temperature stress and damage.

When the finite element model of concrete material is established by the meso-mechanics method, the size and shape of aggregate should be reflected as truly as possible. For the aggregate of three-graded concrete, the maximum particle size is usually 80 mm, and the minimum particle size is 5 mm. Thus, when meshing the model, the maximum size of the element can only be set to a number less than the minimum aggregate particle size of 5 mm. For a dam section with a length of 42 m and a height of about 40 m, tens of millions of elements will be generated after meshing, which will be a great challenge for the computing power of the computer. As described in references [54,55], the meso-scale is a relative scale, and the corresponding scale range can be adjusted according to the specific physical model. When the section of the concrete structure is far larger than the aggregate size, overemphasizing the influence of the aggregate size on the structural stress–strain has no obvious physical significance. The smaller aggregate size can be classified as mortar matrix, while the larger aggregate size can be appropriately adjusted to reflect the heterogeneity of the material in the structural section. Therefore, in this paper, the dam body is discretized into elements with a relatively small size (centimeter order), and the aggregate particles with a small particle size are regarded as the mortar matrix. Considering that the penetrating crack mainly occurs in the upper part of the No. 9 non-overflow dam section, in order to further reduce the calculation scale, the foundation micro-expansive concrete and bedrock are assumed to be homogeneous materials in this calculation, and the macro model is used for simulation. This method of modeling with the meso-mechanics method in the material damage-fracture domain and the macro–mechanics method in the non-damage-fracture domain is called the multi-scale macro–meso model in some of the literature and has proved to be reasonable and feasible [14,55,56]. After modeling by the multi-scale macro–meso method, the mesh size of the meso-part of the model is set to 50 mm, the total number of elements is 448,980, and the total number of nodes is 469,580. In the macro–part of the model, the mesh size is set to 8 m, the total number of elements is 51,058, and the total number of nodes is 20,328.

Following the mesh division, the aggregate is delivered using the approach described in this study. According to Equation (2), the required number of aggregates for two-graded concrete is 30,140, and the required number of aggregates for three-graded concrete is 43,960. The finite element mesh diagram of the concrete aggregate element and ITZ element in the meso-scale is presented in Figure 7, and the overall finite element model of the dam in the macro–meso scale is shown in Figure 8.

In addition, in order to verify the effectiveness of the proposed macro–meso model and the improved anisotropic temperature damage model, the macro-scale model of dam concrete and bedrock is established, and the bilinear damage model proposed in reference [57] is used to calculate and analyze the dam damage. When using the macro scale model, mesh sensitivity analysis shows that different element mesh sizes have no significant effect on damage results. Considering comprehensively such factors as calculation time and space occupied by the hard disk, this paper chooses a mesh size of 3 m for the elements of the dam body and 8 m for the elements of the dam foundation. After mesh partition, the total number of elements is 15,652 and the total number of nodes is 17,910. The overall finite element mesh of No. 9 non-overflow dam section in the macro scale is shown in Figure 9.

### 5.3. Boundary Conditions and Calculation Working Conditions

The Shimantan dam reconstruction project has been in operation for more than 10 years. According to the temperature monitoring data, the concrete hydration process has been basically completed, and the dam temperature field has tended to be stable. It is mainly affected by boundary temperature conditions such as water temperature, air temperature, and ground temperature, showing periodic fluctuations. For the calculation of the boundary conditions of the temperature field, the following assumptions are made based on the temperature monitoring data from the dam, bedrock, and reservoir water: The reservoir water temperature at the equivalent elevation is taken as the upstream dam surface temperature; the air temperature and the solar radiation temperature rise are used to calculate the downstream dam surface temperature; the temperature in the transverse joint is calculated according to the observation data; the temperature of the foundation 15 m below the surface does not fluctuate while the water and air temperatures change, hence a constant value (observation mean value) is used. The specific temperature field calculation boundary conditions are shown in Figure 10.

When calculating the temperature stress and damage, the *x*-direction displacement component is constrained on the upstream and downstream boundaries of the bedrock; the *z*-direction displacement component is constrained on the boundary of both sides of the bedrock along the dam axis; among the 6 displacement components of the nodes on the bottom boundary surface of the model, only 3 linear displacement components are constrained, and the rest are free; due to the action of transverse joints, the joints on the axial boundary of the dam body are unconstrained.

According to the results of on-site safety inspection and the analysis of monitoring data, there is no obvious uneven settlement in the dam foundation, and the dam cracks are closely related to the temperature change.

According to the results of the on-site safety inspection and the analysis of monitoring data, there is no obvious uneven settlement in the dam foundation, and the dam cracks are closely related to the temperature change. Therefore, it is speculated that the cracks in the dam body along the water flow direction should be mainly caused by temperature changes. Based on this, this paper calculates and analyzes the damage to the dam body under the two conditions of temperature rise and temperature drop. According to meteorological conditions in the project area [56], the Shimantan dam site area experienced one cold wave period in December 2002. It suffered from very high temperatures in June 2011, with the peak temperature exceeding 40 °C. The sudden rise and fall of air temperature usually has a great influence on the temperature field of a concrete dam and produces a non-linear temperature difference [53]. The monitoring data indicate that the operating water level of the Shimantan Reservoir is maintained at 105–107 m (normal water level) all year round [58].

Therefore, two types of working conditions, “normal water level + sudden temperature drop” and “normal water level + sudden temperature rise”, are selected for calculation. The duration of temperature drop and temperature rise are selected according to the actual situation. The air temperature and water temperature boundary conditions corresponding to a cold wave period in December 2002 are selected to calculate the cooling temperature field under the temperature drop condition. During this cold wave period, the temperature in the dam site area is reduced from 13 °C to −11 °C within 4 days, so the temperature drop duration is selected as 4 days, and the daily cooling range is −6 °C. The air and water temperature boundary conditions corresponding to a rapid temperature rise process in June 2011 are selected to calculate the temperature rise field under the temperature rise condition. During this rapid temperature rise period, the temperature in the dam site area rises from 21 °C to 40 °C within 3 days, so the temperature rise duration is selected as 3 days, and the daily temperature rise range reaches 6 °C.

Since the main purpose of this paper is to study the influence of temperature change on the macro cracks of the dam, the temperature effect of the dam body is considered separately without considering the effects of mechanical loads such as water pressure and self-weight of the dam body. In this paper, “temperature drop” load refers to the temperature load generated by the difference between the quasi-stable temperature field of the dam body formed in the course of a cold wave and the stable temperature field under the boundary condition of annual average temperature. “Temperature rise” load refers to the temperature load generated by the difference between the quasi-stable temperature field of the dam body formed during the rapid temperature rise process and the stable temperature field under the boundary condition of annual average temperature.

### 5.4. Calculation Parameters

Because the tensile strength of aggregate is significantly greater than that of mortar and ITZ, it is assumed that the aggregate is an elastic material without damage when utilizing the multi-scale macro–meso approach to compute the meso-mechanics properties of concrete. Furthermore, through cracks are mostly found in the upper region of the dam body. As a result, this paper assumes that the foundation micro-expansion concrete and bedrock are elastic materials with no damage. According to the design and construction data and with reference to the relevant literature [10,43,59,60], the proposed material parameters are shown in Table 3 and Table 4.

When using the macroscopic mechanical method and bilinear damage model to compute dam damage, it is assumed that the bedrock is an elastic material that does not produce damage but that the concrete components at all levels of the dam body are damaged. The corresponding parameters are provided in Table 5 based on the design and construction data and references to the relevant literature [53,57,61].

### 5.5. ANSYS Model Establishment and Solution

When the macro–meso model is used for simulation calculation, the information such as aggregate placement boundary and aggregate particle number should be determined first. Then the center coordinate data and radius data of aggregates are randomly generated by the RAND command in the APDL language and stored in a 2D array agv(num, i). In the process of determining material properties, select the geometric entity model of bedrock and foundation micro-expansion concrete, assign corresponding material numbers, select the geometric entity model of the dam body, and set its material type as mortar first. After grid partitioning of the dam model, read the center coordinate data and corresponding radius data of one aggregate, define a local coordinate system at this coordinate point with the LOCAL command, select the element set within this radius with the ESEL command, and change the material type of all elements within this element set from the original defined mortar type to aggregate type. Then one aggregate particle is generated. According to the above steps, the placement of all aggregates can be completed by using the looping statements. Then select all elements connected with each aggregate and change the material type of these elements from the originally defined mortar type to the ITZ type. The ITZ material is selected, and the remaining elements do not change their material properties. These elements are the mortar material of the dam concrete. When all the element attributes of the macro–meso parts of the model are determined, the thermal-structural coupling analysis module of ANSYS software is used to calculate and analyze the temperature field and stress field of the dam. In this paper, the indirect coupling method is used; that is, the temperature field is calculated first, and then the calculation results of the temperature field are extracted. On this basis, the stress field and the damage field are calculated. During temperature field calculation, the solid70 element is selected as the element type. When the temperature field calculation is completed and the stress field calculation is started, the system will automatically convert the element type to the solid185 element. Since the boundary conditions used in the temperature field calculation are different from those required in the stress field calculation, when the temperature field calculation is converted to the stress field calculation, all the boundary conditions in the temperature field calculation need to be deleted and the stress field boundary conditions applied, and the dam stress–strain calculation and analysis can be completed without redividing the element mesh. In addition, it should be noted that the material parameters required for temperature field calculation are different from those for stress field calculation. Therefore, when converting from temperature field to stress field calculation, it is also necessary to redefine the material parameters of each material. When the number of elements and the material properties of elements are unchanged, the program will automatically give the redefined material parameters to each material. In the calculation of structural stress, the anisotropic temperature damage model proposed in this paper is used for the mortar phase and ITZ phase materials of concrete, so it needs to be redeveloped with the help of the user material subroutine USERMAT provided by ANSYS software. The USERMAT subroutine in this paper is programmed and implemented on the Microsoft Visual Studio 2010 + Intel Parallel Studio XE 2011 platform, and then compiled and linked to the user-customized version of ANSYS. exe executable file on the ansys15.0 (64 bit) platform. In the structural calculation, the stress and damage distribution of the dam concrete can be calculated according to the anisotropic temperature damage model proposed in this paper by calling this version of ANSYS.exe. In the macro model calculation, there is no need to add aggregate, and other calculation processes are the same as those of the macro–meso model. For the bilinear damage model used in macro model calculation, it also needs to be realized by secondary development with the help of user material subroutine USERMAT. Its development platform is the same as the above, and will not be repeated here. The specific flow chart of macro–meso model calculation is shown in Figure 11, and the flow chart of macro model calculation is shown in Figure 12.

### 5.6. Analysis of Calculation Results

#### 5.6.1. Calculation RESULTS of Macro–Meso Model

##### Normal Water Level + Sudden Temperature Drop

Under the working condition of “normal water level + sudden temperature drop”, the cloud charts of the overall temperature damage of the dam in three principal strain directions calculated by the macro–meso model in this paper are shown in Figure 13.

In sction z = 21 m along the axis direction of the dam, the temperature damage cloud charts in the three principal strain directions are shown in Figure 14.

The temperature damage cloud charts of the mortar material of dam concrete in the three principal strain directions are shown in Figure 15.

The temperature damage cloud charts of the ITZ material of dam concrete in the three principal strain directions are shown in Figure 16.

The three principal strain cloud charts on the section z = 21 m along the dam axis are shown in Figure 17.

##### Normal Water Level + Sudden Temperature Rise

Under the working condition of “normal water level + sudden temperature rise”, the cloud charts of the overall temperature damage of the dam in the three principal strain directions calculated by the macro–meso model in this paper are shown in Figure 18.

In section z = 21 m along the axis direction of the dam, the temperature damage cloud charts in the three principal strain directions are shown in Figure 19.

The temperature damage cloud charts of the mortar material of dam concrete in the three principal strain directions are shown in Figure 20.

The temperature damage cloud charts of the ITZ material of dam concrete in the three principal strain directions are shown in Figure 21.

The three principal strain cloud charts on the section z = 21 m along the dam axis are shown in Figure 22.

##### Result Analysis

It can be seen from Figure 13, Figure 14, Figure 15 and Figure 16 that under the working condition of “normal water storage + sudden temperature drop”, the damage in the three principal strain directions caused by the sudden temperature drop mainly occurs in the mortar area and ITZ area of the concrete on the upstream and downstream surfaces of the dam body, while there is no damage inside the dam body, which conforms to the general law of crack distribution on the surface of concrete dams during the sudden temperature drop. When the temperature in the dam site area drops suddenly, the concrete temperature on the surface of the dam body decreases rapidly, and the surface concrete immediately shrinks and deforms, while the internal concrete temperature changes slowly and the volume changes little. The shrinkage deformation of the surface concrete is constrained by the internal concrete, resulting in a large tensile strain on the surface of the dam body and a compressive strain on the internal concrete (see Figure 17). According to the definition of the model in this paper, the tensile strain produces damage, and the compressive strain does not produce damage. From the calculation results, the distribution of damage in the three main strain directions is consistent with the strain distribution, and the calculation results of temperature damage conform to the expectations of the model in this paper.

It can be seen from Figure 18, Figure 19, Figure 20 and Figure 21 that under the working condition of “normal water storage + sudden temperature rise”, the sudden temperature rise can not only cause minor damage on the upstream and downstream surfaces of the dam but also cause minor damage in the mortar area and ITZ area of the concrete inside the dam. When the temperature in the dam site rises suddenly, the temperature of the concrete on the surface of the dam body rises rapidly, and the surface concrete immediately expands and deforms, while the temperature of the internal concrete changes slowly and the volume changes little. The expansion and deformation of the external concrete are constrained by the internal concrete, so that the internal concrete will produce a certain degree of tensile strain due to the tensile action of the expansion deformation of the external concrete (see Figure 22). In addition, since the thermal expansion coefficient of the aggregate material is smaller than that of mortar and ITZ materials, the thermal expansion deformation of the mortar phase and ITZ phase of surface concrete is larger, while the deformation of aggregate is relatively small. The aggregate will restrict the expansion deformation of the mortar and ITZ, resulting in a certain tensile strain on the mortar phase and ITZ phase of the surface concrete. As a result, the dam body will experience some degree of damage from the sudden temperature rise under the working condition of “normal water storage + sudden temperature rise”, but the damage amount is minimal.

#### 5.6.2. Calculation Results of Macro Model

##### Normal Water Level + Sudden Temperature Drop

Under the condition that the applied boundary conditions and temperature loads are exactly the same as those of the macro–meso model, the damage of the dam body under the working condition of “normal water level + sudden temperature drop” calculated by the macro model is shown in Figure 23.

The three principal strain cloud charts on the section z = 21 m along the dam axis are shown in Figure 24.

##### Normal Water Level + Sudden Temperature Rise

Under the condition that the applied boundary conditions and temperature loads are exactly the same as those of the macro–meso model, the damage of the dam body under the working condition of “normal water level + sudden temperature rise” calculated by the macro model is shown in Figure 25.

The three principal strain cloud charts on the section z = 21 m along the dam axis are shown in Figure 26.

##### Result Analysis

It can be seen from Figure 23 that under the working condition of “normal water level + sudden temperature drop”, a certain degree of damage will occur at the surface concrete of the downstream dam body contacting air, with the maximum damage value reaching 0.76. However, there is no damage in the large area inside the dam body, which conforms to the general law of crack distribution on the surface of the dam body under the condition of a sudden temperature drop. From Figure 24, it can be seen that the tensile strain is mainly generated on the upstream and downstream surfaces of the dam body, and the internal part of the dam body is mainly compressive strain, which also conforms to the general law of strain distribution of the dam body under the condition of temperature drop. It can be seen from Figure 25 that under the condition of “normal water storage + sudden temperature rise”, a certain degree of damage will occur inside the dam body at the middle part of the downstream dam, and a certain degree of damage will also occur at the toe of the dam. The maximum damage value is 0.17. The reason for the damage of the concrete below the surface layer in the middle part of the downstream dam may be that the temperature rises suddenly, and the concrete in this part expands and deforms when it is heated, while the temperature of the internal concrete in the depth of the dam changes little, which restricts the expansion and deformation of the concrete in this part. In addition, this part is located at the junction of the three-graded concrete and the two-graded concrete. The thermal expansion coefficient of the three-graded concrete is slightly larger than that of the two-graded concrete, and the two-graded concrete is below the downstream water level. The influence of the sudden temperature rise on the two-graded concrete is smaller than that of the three-graded concrete, and the volume change is relatively small. After the three-graded concrete is heated and expanded, it cannot deform vertically downward due to the restraint of the two-graded concrete. It can only deform downstream along the junction of the two-graded concrete and the three-graded concrete, and under the constraint of the concrete inside the dam, the three-graded concrete cannot deform downstream freely. Therefore, under the joint constraint of the concrete inside the dam and the two-graded concrete, the concrete below the surface layer in the middle of the downstream dam produces a large tensile strain (see Figure 26a). According to the calculation results, one can get that under the calculation conditions of this paper, the damage distribution of the dam body under the working condition of “normal water storage + sudden temperature rise” calculated by the macro model and bilinear damage model is not very consistent with the actual situation that there are two through cracks penetrating the upstream and downstream of the dam section.

#### 5.6.3. Comparative Analysis of Calculation Results

Based on the damage calculation results, under the working condition of “normal water storage + sudden temperature drop”, the macro–meso model of the dam is constructed by using the aggregate placement method proposed in this paper, and when combined with the improved anisotropic temperature damage model in this paper, the calculated damage distribution of No. 9 non-overflow dam section of Shimantan dam conforms to the general law of concrete crack distribution on the surface of the dam body under the condition of sudden temperature drop. The damage distribution of the dam body calculated by the macro model combined with the bilinear damage model also conforms to the general law of the crack distribution of the dam surface concrete under the temperature drop condition. However, under the working condition of “normal water storage + sudden temperature rise”, the distribution of temperature damage calculated by the two methods is quite different. The damage distribution area calculated by the macro–meso model combined with the improved anisotropic temperature damage model in this paper is consistent with the distribution area of tensile strain, while the damage area calculated by the macro model combined with the bilinear damage model is quite different from the distribution area of tensile strain, and there is also a certain degree of damage in the compressive strain area. There are two through cracks in the No. 9 dam section of Shimantan dam that penetrate the upstream face, dam crest, and downstream face and are perpendicular to the dam axis. However, the results calculated by the macro model combined with the bilinear damage model indicate that the dam may produce longitudinal joints parallel to the dam axis in the middle of the downstream dam surface, which is not very consistent with the actual crack distribution.From the strain calculation results, when the macro model is used and combined with the bilinear damage model, the material partition is mainly completed by giving the corresponding material parameters to different dam blocks and bedrock. Due to the different elastic modulus and thermal conductivity of different materials, the deformation at the interface of different materials will be inconsistent, resulting in excessive strain gradient, Thus, the damage calculation results are inconsistent with the actual situation. When the macro–meso model is used and combined with the improved anisotropic temperature damage model in this paper, the material partition of dam concrete is mainly reflected by changing the aggregate content of the two- and three-graded concrete, while the material parameters are assigned to the aggregate, mortar, and ITZ respectively. Due to the randomness of aggregate distribution, there will be no excessive strain gradient at the interface of two- and three-graded concrete due to the different material properties of the two types of concrete, which will affect the calculation results of damage.From the perspective of the causes of the two through cracks in the No. 9 dam section of Shimantan dam, under the working condition of “normal water storage + sudden temperature drop”, large tensile stress and damage are generated on the dam surface calculated by the above two methods. From the perspective of damage, it is judged that the upstream and downstream surfaces of the dam body will produce macro cracks, which is consistent with the traditional strength theory; that is, when the temperature drops suddenly, it will lead to excessive tensile stress in the concrete on the dam surface, which leads to the generation of macroscopic cracks on the surface. Under the working condition of “normal water storage + sudden temperature rise”, small tensile stress and damage are generated in the dam body as calculated by the above two methods. From the perspective of traditional strength theory, since the tensile stress in the dam body does not exceed the tensile strength of concrete, the sudden temperature rise has little effect on the formation of cracks in concrete dams. However, from the perspective of damage mechanics, the formation of macro cracks is the result of the gradual accumulation of damage, and the damage of concrete materials is irreversible. Although the internal damage of the dam body caused by a rapid temperature rise process is small, if the dam experiences multiple sudden temperature rises, the damage inside the dam body may gradually accumulate until the dam forms macro cracks and finally fractures. It can be seen from the meteorological data that although the dam site area of Shimantan Dam belongs to the temperate monsoon climate, weather with rapid cooling in winter and rapid heating in summer often occurs. Based on the above analysis, this paper holds that under the condition that there is no obvious uneven settlement in the dam foundation, the cause of the two through cracks in the No. 9 non-overflow dam section of Shimantan dam is mainly due to the formation of surface macro cracks on the dam surface after the dam experiences a cold wave and rapid cooling, while after the next cold wave, the surface cracks develop towards the inside of the dam. However, the internal micro damage of the dam caused by the sudden temperature rise in summer leads to the deterioration of the performance of concrete materials. Under the intertwined influence of the sudden temperature drop, the internal micro damage continues to develop. After several cold and hot cycles of sudden temperature drops in winter and sudden temperature rises in summer, the internal damage of the dam gradually accumulates until the final formation of macro penetrating cracks.

According to the above comparative analysis, for the analysis of the causes of cracks in concrete dams, one cannot arbitrarily hold that the formation of macro cracks in concrete dams is due to the excessive tensile stress caused by the sudden temperature drop, and the sudden temperature rise has no effect on the formation of macro cracks in concrete dams. From the perspective of damage mechanics, even if the tensile stress does not exceed the tensile strength of the material, as long as there is tensile strain, damage may occur in the material. From the perspective of fracture mechanics, for the structure with initial cracks, when the working stress of the structure is far less than the yield strength of the material, low-stress brittle fracture may also occur in the material, and this kind of low-stress brittle fracture accident also occurs from time to time in engineering. According to the above comparative analysis, it can also be seen that the damage distribution of the dam body calculated by using the macro–meso model, combined with the improved anisotropic temperature damage model proposed by this paper, is more in accord with the actual situation of the crack distribution of the No. 9 non-overflow dam section of Shimantan dam. Moreover, the rationality and feasibility of the macro–meso model and the anisotropic temperature damage model proposed in this paper are further verified.

## 6. Conclusions

In this paper, a new method of generating concrete random aggregate models is proposed by using the APDL language of ANSYS software. Moreover, an improved anisotropic temperature damage model, which can take into account the interaction between aggregates, ITZ, and mortar, is proposed. By using this new aggregate placement method, the macro–meso finite element model of the No. 9 non-overflow dam section of the Shimantan dam has been constructed. On this basis, combined with the anisotropic concrete temperature damage model, the temperature damage of the No. 9 non-overflow dam section of the Shimantan dam is calculated and analyzed. The simulation results of the aggregate placement effectiveness of small-scale concrete specimens and the finite element numerical simulation results of the temperature damage of Shimantan dam show that:The 2D circular, polygonal, and elliptical random concrete aggregate models can be quickly generated by using the aggregate placement method proposed in this paper. For 3D problems, spherical, polyhedral, and ellipsoidal random aggregates can be generated in a very short time, which greatly improves the efficiency of random aggregate placement. Moreover, the proposed aggregate placement method can meet the modeling requirements of the 3D meso model of the large-scale concrete structure.Compared with the results of the macroscopic finite element model combined with the bilinear damage model, the temperature damage of the No.9 non-overflow dam section of the Shimantan dam calculated by using the macro–meso finite element model and the improved anisotropic temperature damage model is more consistent with the actual distribution of dam cracks, which shows that the proposed macro–meso finite element model and the improved anisotropic temperature damage model are reasonable and feasible. It provides a new way to simulate the temperature damage of mass concrete structures.The calculation results of the No. 9 non-overflow dam section of Shimantan dam show that the periodic rapid temperature drop in winter and the rapid temperature rise in summer are the main reasons for the temperature damage and cracks of Shimantan concrete gravity dam during the operation period. In the future dam operation process, permanent thermal insulation measures should be taken for the dam surface, and running water cooling measures can be taken for the dam surface in the high temperature season in summer.When assessing the structural safety of concrete dams, one should combine damage mechanics theory or fracture mechanics theory for auxiliary analysis and judgment, rather than relying solely on traditional strength theory to determine whether the dam will fracture under a specific load. When considering the effect of temperature load on the dam, besides the concrete damage on the surface of the dam caused by the sudden temperature drop, the internal damage of the dam caused by the rapid rise in temperature cannot be ignored.

## Figures and Tables

**Figure 1 materials-15-07138-f001:**
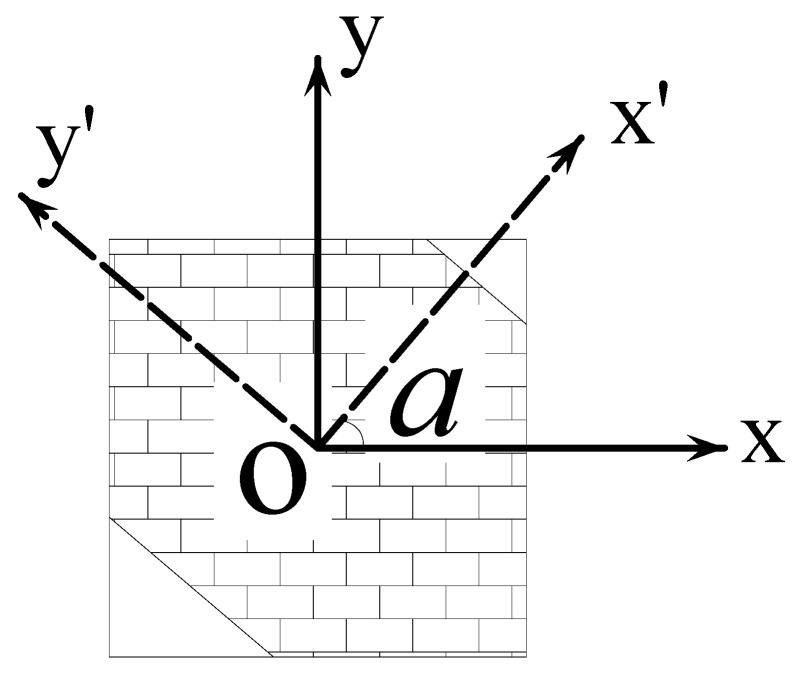
Polygonal aggregate element selection schematic diagram.

**Figure 2 materials-15-07138-f002:**
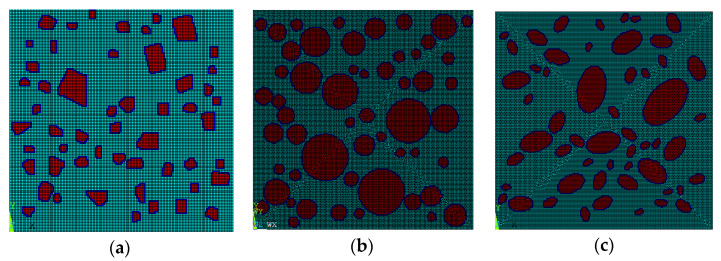
Finite element mesh diagram for 2D random aggregate models, (**a**) Polygonal aggregate, (**b**) Circular aggregate, (**c**) Elliptical aggregate.

**Figure 3 materials-15-07138-f003:**
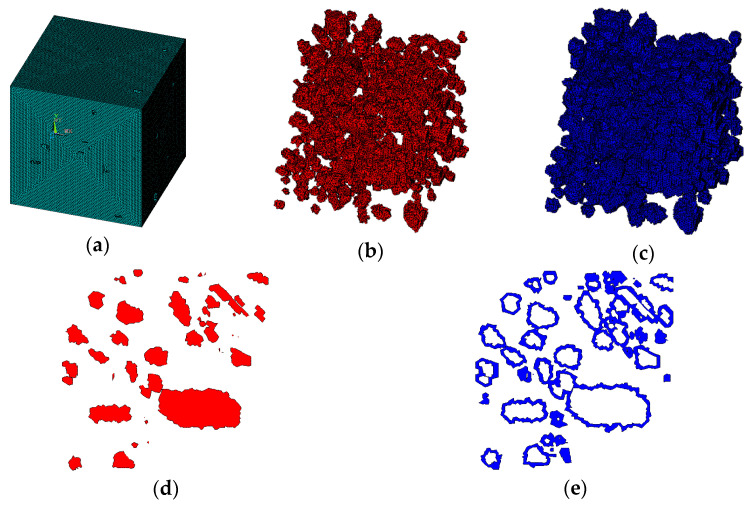
Mesh diagram of mortar, aggregate and ITZ elements in polyhedral aggregate model, (**a**) Mortar elements, (**b**) aggregate elements, (**c**) ITZ elements, (**d**) Aggregate elements on the profile, (**e**) ITZ elements on the profile.

**Figure 4 materials-15-07138-f004:**
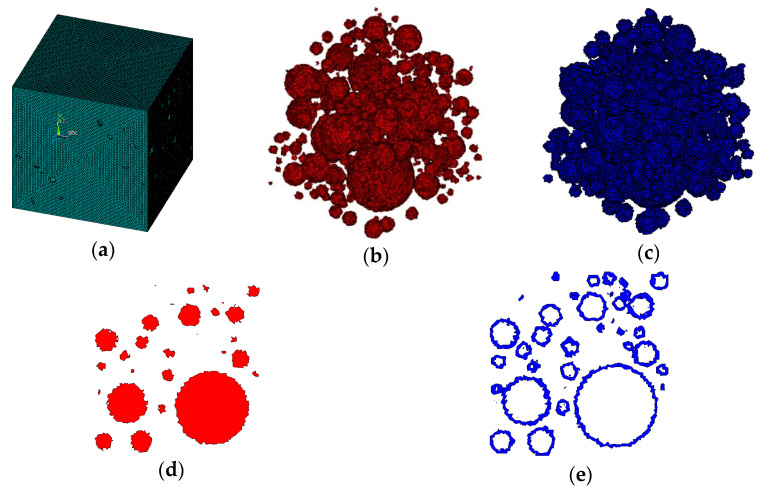
Mesh diagram of mortar, aggregate and ITZ elements in the spherical aggregate model, (**a**) Mortar elements, (**b**) aggregate elements, (**c**) ITZ elements, (**d**) Aggregate elements on the profile, (**e**) ITZ elements on the profile.

**Figure 5 materials-15-07138-f005:**
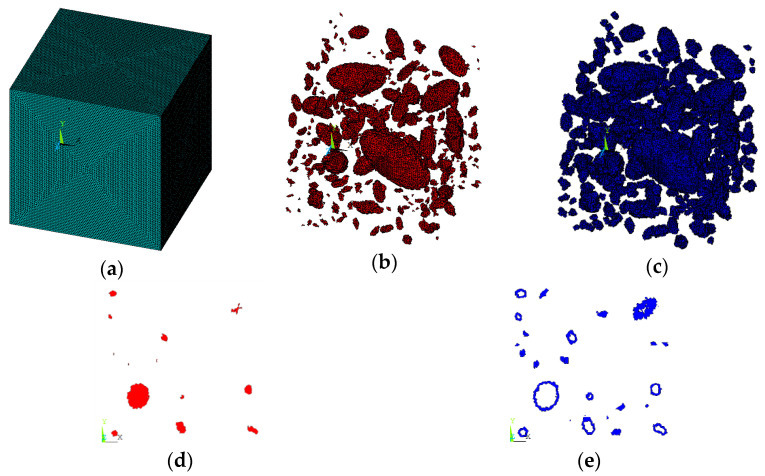
Mesh diagram of mortar, aggregate and ITZ elements in the ellipsoidal aggregate model, (**a**) Mortar elements, (**b**) aggregate elements, (**c**) ITZ elements, (**d**) Aggregate elements on the profile, (**e**) ITZ elements on the profile.

**Figure 6 materials-15-07138-f006:**
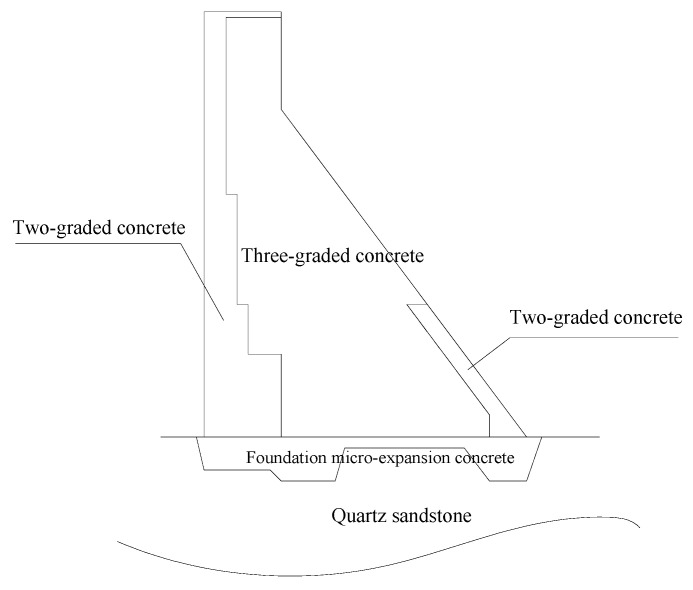
Section sketch and material partition diagram of No. 9 non-overflow dam section.

**Figure 7 materials-15-07138-f007:**
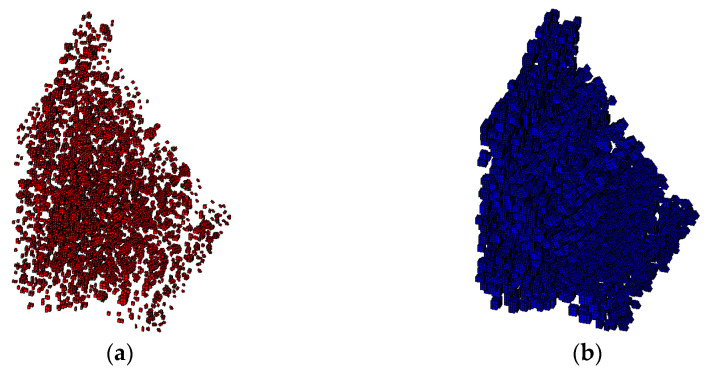
Finite element mesh diagram of dam concrete aggregate and ITZ elements, (**a**) Aggregate elements, (**b**) ITZ elements.

**Figure 8 materials-15-07138-f008:**
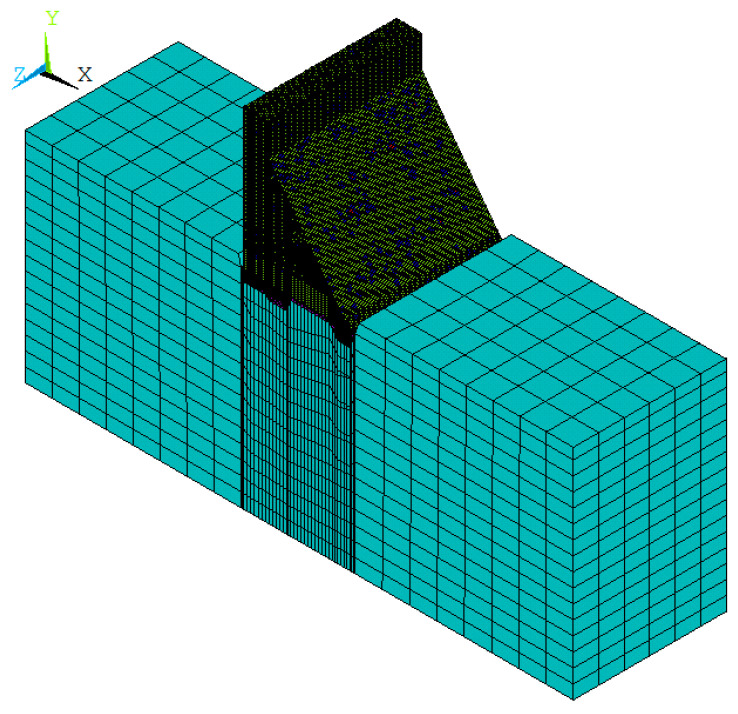
Overall finite element model of the dam in the macro–meso scale.

**Figure 9 materials-15-07138-f009:**
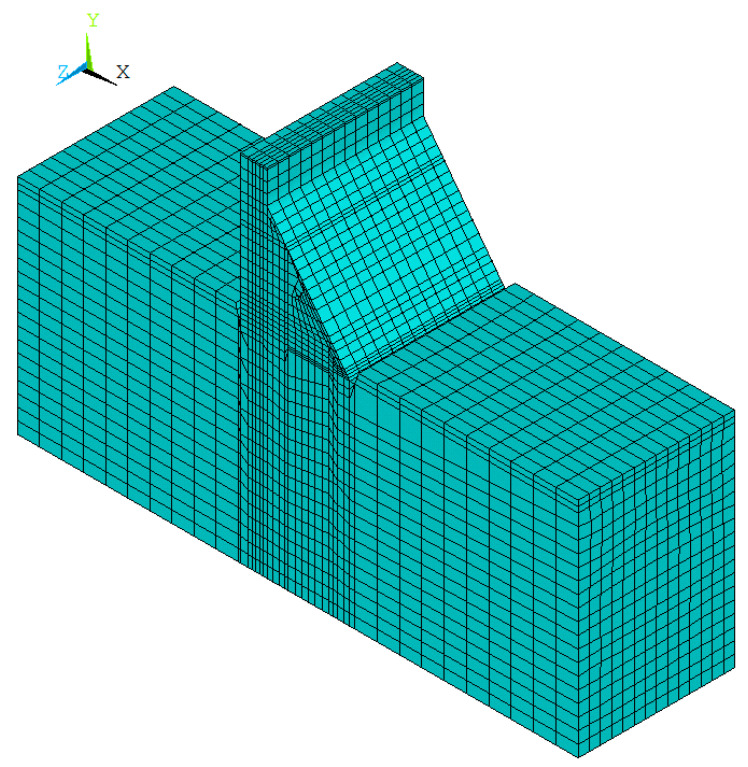
Overall finite element mesh of the dam in the macro scale.

**Figure 10 materials-15-07138-f010:**
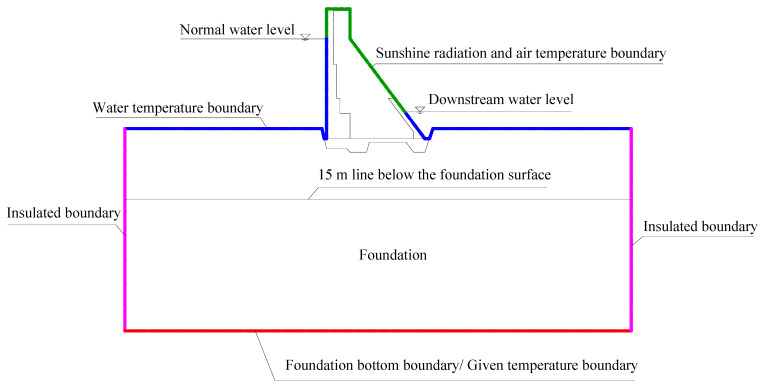
Boundary conditions for the temperature field calculation.

**Figure 11 materials-15-07138-f011:**
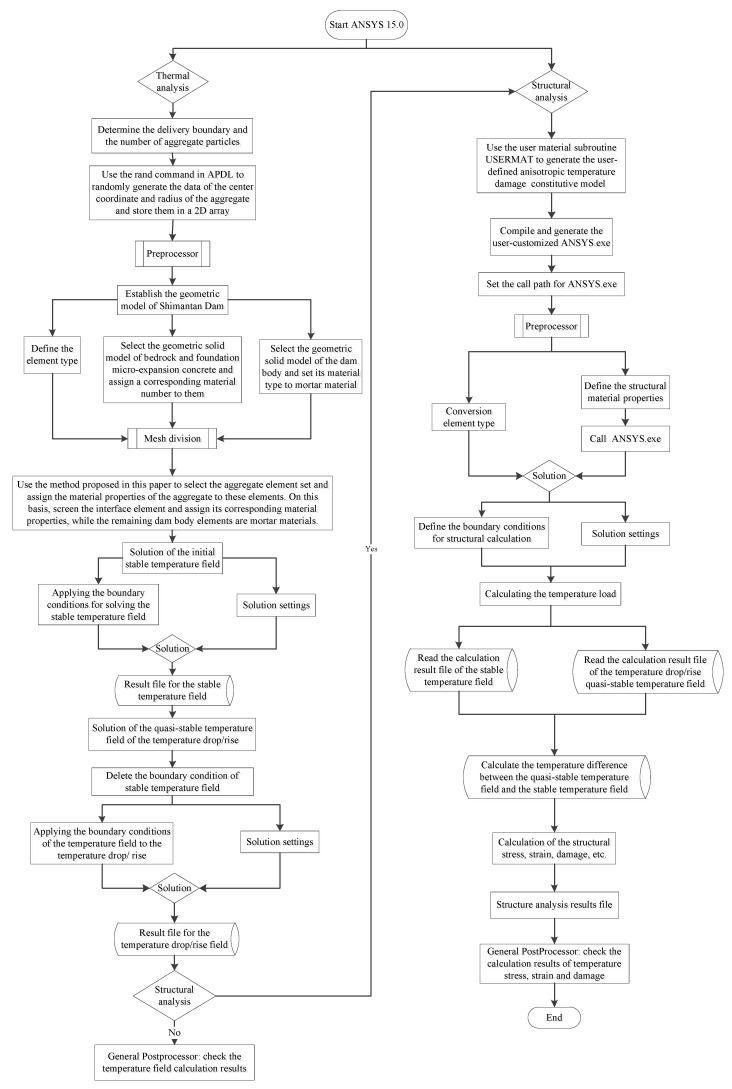
The flow chart of macro–meso model calculation.

**Figure 12 materials-15-07138-f012:**
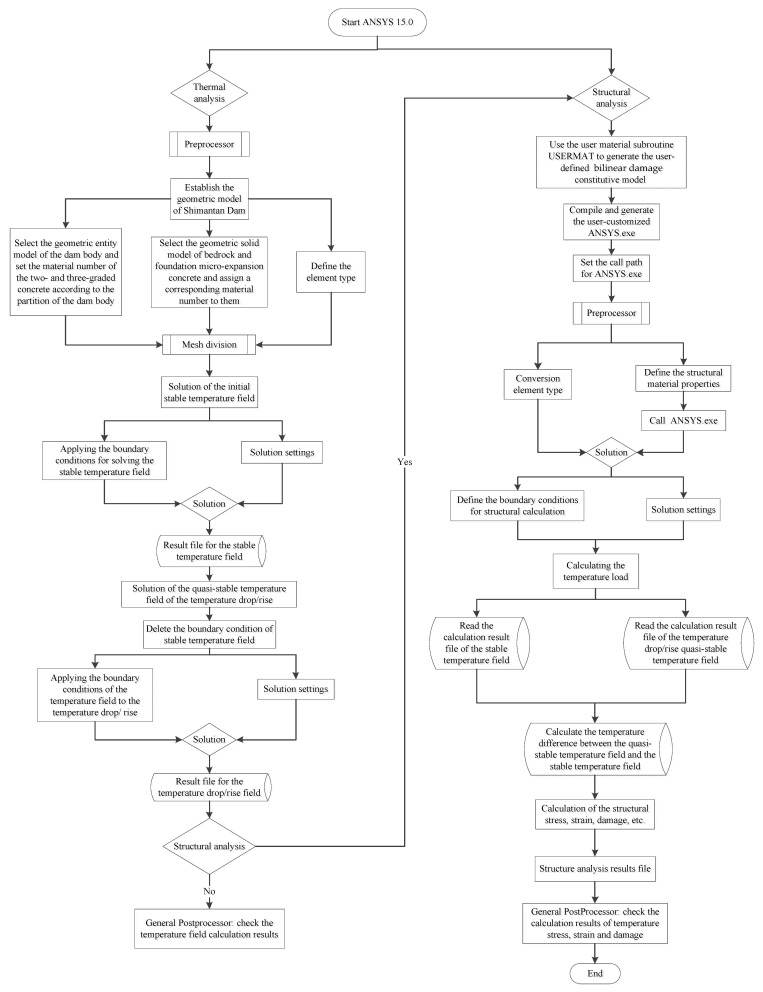
The flow chart of macro model calculation.

**Figure 13 materials-15-07138-f013:**
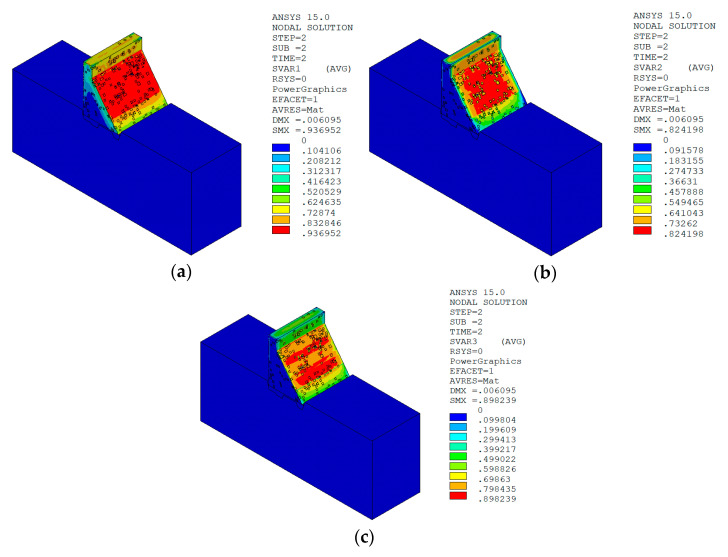
The temperature damage distribution of the dam body in the three principal strain directions is calculated by the macro-meso model under the temperature drop condition, (**a**) The temperature damage distribution in the first principal strain direction, (**b**) The temperature damage distribution in the second principal strain direction, (**c**) The temperature damage distribution in the third principal strain direction.

**Figure 14 materials-15-07138-f014:**
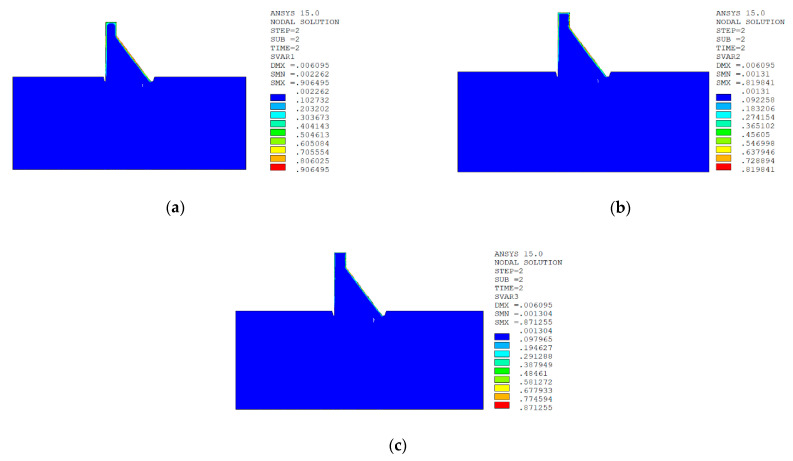
The temperature damage distribution in the three principal strain directions on the section z = 21 m is calculated by the macro-meso model under the temperature drop condition, (**a**) The temperature damage distribution in the first principal strain direction, (**b**) The temperature damage distribution in the second principal strain direction, (**c**) The temperature damage distribution in the third principal strain direction.

**Figure 15 materials-15-07138-f015:**
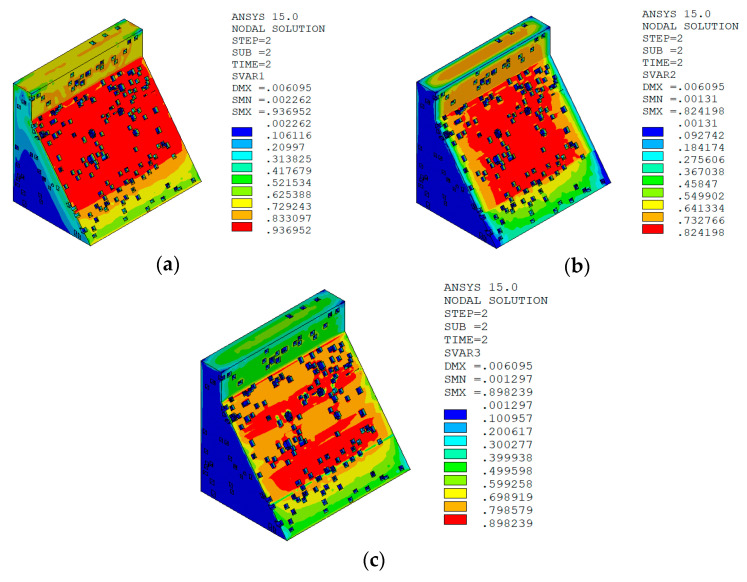
The temperature damage distribution of the mortar material in the three principal strain directions is calculated by the macro-meso model under the temperature drop condition, (**a**) The temperature damage distribution in the first principal strain direction, (**b**) The temperature damage distribution in the second principal strain direction, (**c**) The temperature damage distribution in the third principal strain direction.

**Figure 16 materials-15-07138-f016:**
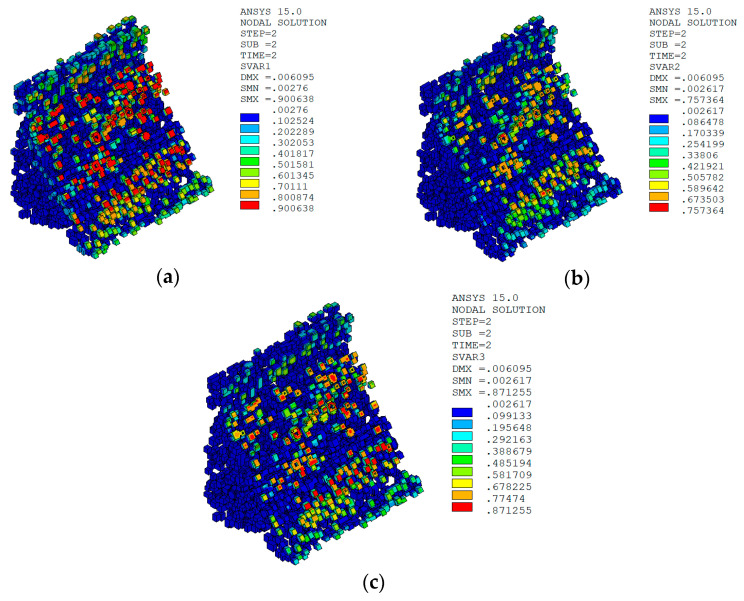
The temperature damage distribution of the ITZ material in the three principal strain directions is calculated by the macro-meso model under the temperature drop condition, (**a**) The temperature damage distribution in the first principal strain direction, (**b**) The temperature damage distribution in the second principal strain direction, (**c**) The temperature damage distribution in the third principal strain direction.

**Figure 17 materials-15-07138-f017:**
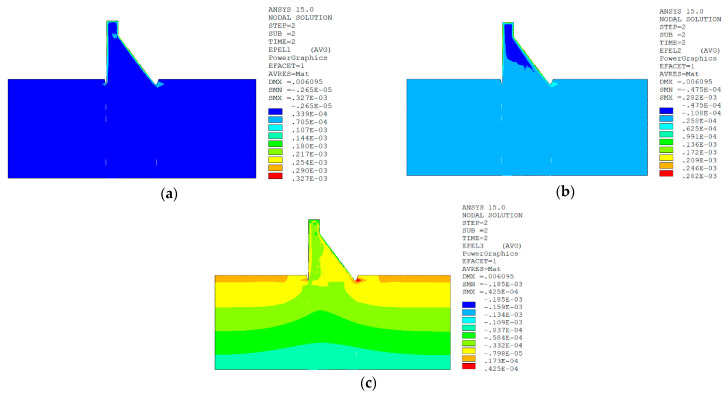
The cloud charts of the three principal strains on the section z = 21 m are calculated by the macro-meso model under the temperature drop condition, (**a**) The cloud chart of the first principal strain, (**b**) The cloud chart of the second principal strain, (**c**) The cloud chart of the third principal strain.

**Figure 18 materials-15-07138-f018:**
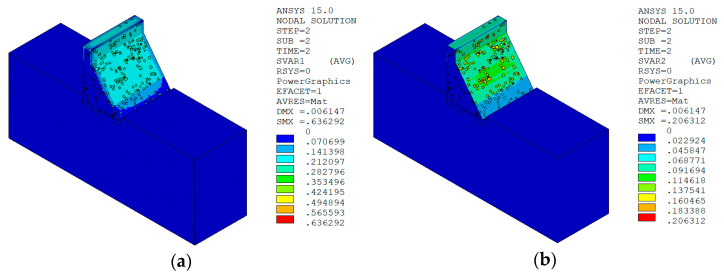
The temperature damage distribution of the dam body in the three principal strain directions is calculated by the macro-meso model under the temperature rise condition, (**a**) The temperature damage distribution in the first principal strain direction, (**b**) The temperature damage distribution in the second principal strain direction, (**c**) The temperature damage distribution in the third principal strain direction.

**Figure 19 materials-15-07138-f019:**
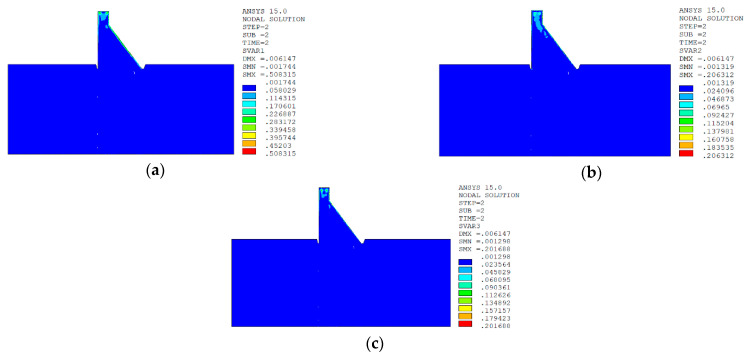
The temperature damage distribution in the three principal strain directions on the section z = 21 m is calculated by the macro-meso model under the temperature rise condition, (**a**) The temperature damage distribution in the first principal strain direction, (**b**) The temperature damage distribution in the second principal strain direction, (**c**) The temperature damage distribution in the third principal strain direction.

**Figure 20 materials-15-07138-f020:**
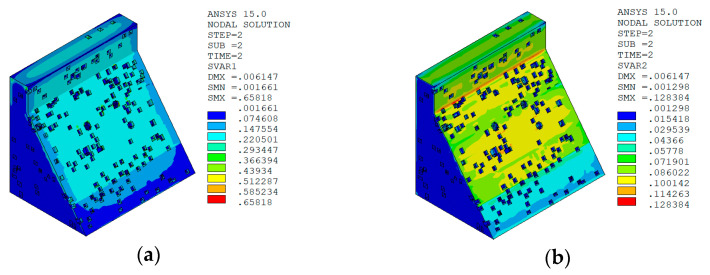
The temperature damage distribution of the mortar material in the three principal strain directions is calculated by the macro-meso model under the temperature rise condition, (**a**) The temperature damage distribution in the first principal strain direction, (**b**) The temperature damage distribution in the second principal strain direction, (**c**) The temperature damage distribution in the third principal strain direction.

**Figure 21 materials-15-07138-f021:**
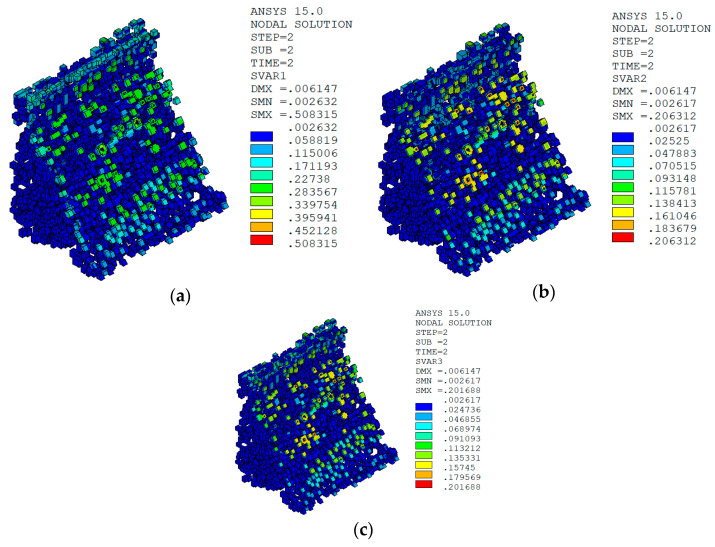
The temperature damage distribution of the ITZ material in the three principal strain directions is calculated by the macro-meso model under the temperature rise condition, (**a**) The temperature damage distribution in the first principal strain direction, (**b**) The temperature damage distribution in the second principal strain direction, (**c**) The temperature damage distribution in the third principal strain direction.

**Figure 22 materials-15-07138-f022:**
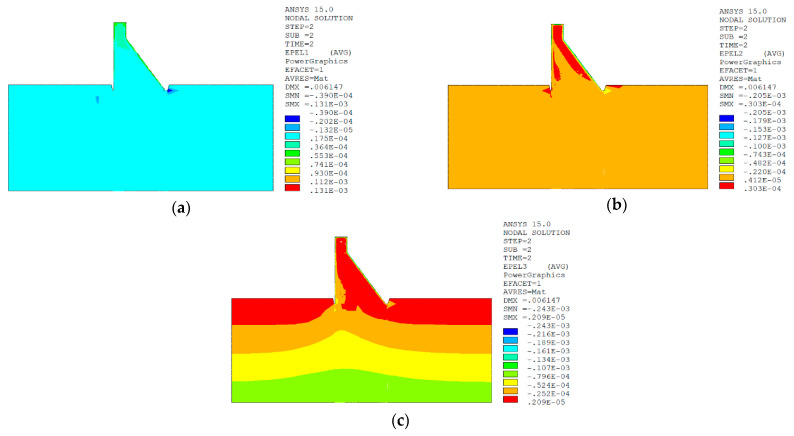
The cloud charts of the three principal strains on the section z = 21 m are calculated by the macro-meso model under the temperature rise condition, (**a**) The cloud chart of the first principal strain, (**b**) The cloud chart of the second principal strain, (**c**) The cloud chart of the third principal strain.

**Figure 23 materials-15-07138-f023:**
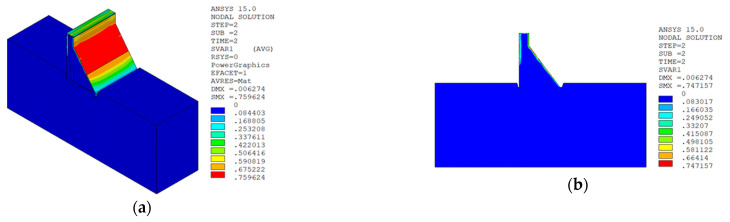
The temperature damage distribution of the dam is calculated by the macro model under the temperature drop condition, (**a**) Nephogram of temperature damage distribution of the dam, (**b**) Nephogram of temperature damage distribution of section z = 21 m.

**Figure 24 materials-15-07138-f024:**
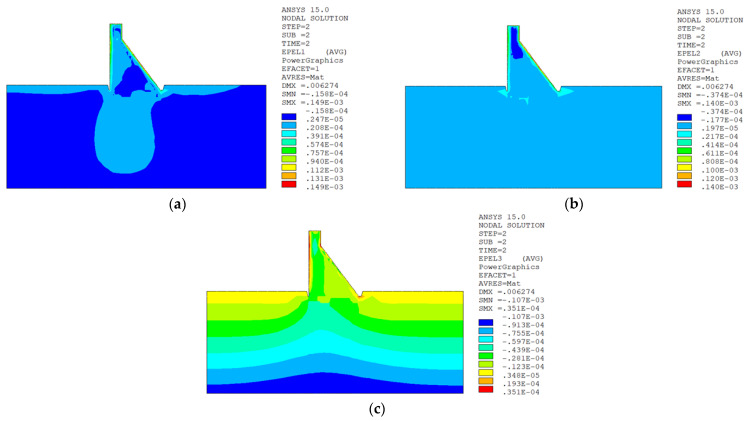
The cloud charts of the three principal strains on the section z = 21 m are calculated by the macro model under the temperature drop condition, (**a**) The cloud chart of the first principal strain, (**b**) The cloud chart of the second principal strain, (**c**) The cloud chart of the third principal strain.

**Figure 25 materials-15-07138-f025:**
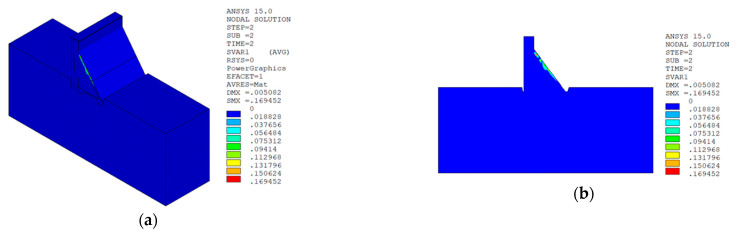
The temperature damage distribution of the dam is calculated by the macro model under the temperature rise condition, (**a**) Nephogram of temperature damage distribution of the dam, (**b**) Nephogram of temperature damage distribution of section z = 21 m.

**Figure 26 materials-15-07138-f026:**
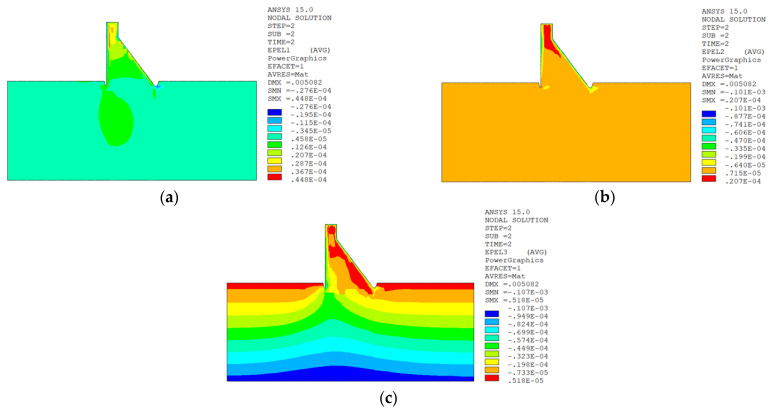
The cloud charts of the three principal strains on the section z = 21 m are calculated by the macro model under the temperature rise condition, (**a**) The cloud chart of the first principal strain, (**b**) The cloud chart of the second principal strain, (**c**) The cloud chart of the third principal strain.

**Table 1 materials-15-07138-t001:** Distribution probability of aggregate for each particle size.

*d*_0_/mm	40	20	5
*d* _0_ */d* _max_	1	0.5	0.125
*P_c_*	0.645	0.487	0.230

Note: *d*_0_ indicates aggregate particle size; *d*_0_*/d*_max_ indicates the ratio of aggregate particle size to maximum aggregate particle size; *P_c_* indicates particle size less than *d*_0_ ratio of the aggregate area of 0 to the cross-sectional area of the specimen.

**Table 2 materials-15-07138-t002:** Ratio of specimen area to aggregate area.

Aggregate Radius/mm	15	6.25
Single aggregate area/mm^2^	706.5	122.66
*A/A_i_*	31.85	183.43

Note: *A/A_i_* represents the ratio of the cross-sectional area of the specimen to the area of a single aggregate.

**Table 3 materials-15-07138-t003:** Meso material parameters of multiscale macro–meso model.

Parameter	Sign	Unit	Material		
			Mortar	Aggregate	ITZ
Density	ρ	Kg/m^3^	2151	2700	2360
Young’s Modulus	*E*	GPa	21	60	20
Poisson’s ratio	*ν*	-	0.22	0.167	0.21
Thermal Conductivity	*λ*	kJ/(m▪day▪°C)	121	242	184
Heat Capacity	*c*	kJ/(m▪°C)	0.94	0.77	0.91
Thermal Expansion Coefficient	*α*	10^−6^/°C	10	7	13
Tension Strength	*f*	MPa	1.66	-	1.02
Fracture Energy	*G*	N/mm^2^	143	-	109.2

**Table 4 materials-15-07138-t004:** Material parameters of the macro part of the multi-scale macro–meso model.

Parameter	Sign	Unit	Material	
			Bedrock	Micro-Expansion Concrete
Density	ρ	Kg/m^3^	2540	2400
Young’s Modulus	*E*	GPa	13	25.5
Poisson’s ratio	*ν*	-	0.23	0.167
Thermal Conductivity	*λ*	kJ/(m▪day▪°C)	164.88	187.92
Heat Capacity	*c*	kJ/(m▪°C)	0.77	0.87
Thermal Expansion Coefficient	*α*	10^−6^/°C	8	9

**Table 5 materials-15-07138-t005:** Material parameters of the macro scale model.

Parameter	Sign	Unit	Material			
			Bedrock	Micro-Expansion Concrete	Two-Graded Concrete	Three-Graded Concrete
Density	ρ	Kg/m^3^	2540	2400	2351	2360
Young’s Modulus	*E*	GPa	13	25.5	25.5	22
Poisson’s ratio	*ν*	-	0.23	0.167	0.167	0.167
Thermal Conductivity	*λ*	kJ/(m▪day▪°C)	164.88	187.92	184	184
Heat Capacity	*c*	kJ/(m▪°C)	0.77	0.87	0.94	0.94
Thermal Expansion Coefficient	*α*	10^−6^/°C	7	8	5.8	6.7
Tensile damage threshold strain	*ε* * _tf_ *	-	1.2 × 10^−4^	6.84 × 10^−5^	6.84 × 10^−5^	6.18 × 10^−5^
Tensile damage limit strain	*ε* * _tu_ *	-	1.2 × 10^−3^	6.84 × 10^−4^	6.84 × 10^−4^	6.18 × 10^−4^
Compressive damage threshold strain	*ε* * _cf_ *	-	1.64 × 10^−3^	1.23× 10^−3^	1.23 × 10^−3^	1.16 × 10^−3^
Compressive damage limit strain	*ε* * _cu_ *	-	1.64 × 10^−2^	1.23 × 10^−2^	1.23 × 10^−2^	1.16 × 10^−2^

## Data Availability

Not applicable.

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
