# Peer review of "Calculation and Analysis of Temperature Damage of Shimantan Concrete Gravity Dam Based on Macro–Meso Model"

_materials, 2022, doi:10.3390/ma15207138_

Round 1
Reviewer 1 Report
The authors have made an attempt to develop a model for generating concrete random aggregate to be used in finite element researches. To this aim, APDL language of ANSYS FEA is used to analyze the model on Shimantan dam. The outcomes are compared and discussed and conclusions are made accordingly.
The subject matter is of high importance and the paper is well-organized. The following comments, however, should be addressed by the authors in order to enhance the quality of the manuscript:
1. P4, L185: Eq. 1 presents X and Y, while Z is also required, as also mentioned in Line 185. it is not clear how Z is considered in a random generation process.
2. Section 4: the authors are highly recommended to mention the Concrete Damage Plasticity (CDP) model which is widely used by researchers for modeling concrete behavior in FEA studies. The following references might be helpful:
a. Dabiri, H., Kaviani, A., & Kheyroddin, A. (2020). Influence of reinforcement on the performance of non-seismically detailed RC beam-column joints. Journal of Building Engineering, 31, 101333.
b. Behnam, H., Kuang, J. S., & Samali, B. (2018). Parametric finite element analysis of RC wide beam-column connections. Computers & Structures, 205, 28-44.
3. Fig. 9: mesh size of the model in different parts are not the same. The authors should explain if it can affect the accuracy of the results.
4. Fig. 9: This figure should be enlarged such that the meshing of different parts could be illustrated easily. It seems that the model is not meshed appropriately in some parts (e.g., the soil and dam interface). Moreover, the authors should explain how this mesh size is selected. Have they performed any sensitivity analysis, or the mesh size is chosen based on any recommendations in literature?
5. P20, L745-746: it is recommended that the authors add a figure demonstrating the boundary conditions clearly.
6. Adding a Nomenclature could help readers find the parameters easier and therefore, the authors are recommended to add it to the end of the manuscript.
Reviewer 2 Report
The authors have presented a interesting study on the Calculation and Analysis of Temperature Damage of Shiman-2 tan Concrete Gravity Dam based on Macro-Meso Model.
Based on the detailed overview, the presented study is well written and explains all the required parameters/aspects in a proper way.
The authors addressed all the relevant concerns in the introduction part.
The study briefly explains and discusses all the methodology preciously.
I would accept the presented study for publication in the present form.
Reviewer 3 Report
In the Reviewer opinion the research paper entitled “Calculation and analysis of temperature damage of Shimantan concrete gravity dam based on macro-meso model” is good.
In this study a new method of random aggregate delivery using the ESEL command in APDL and the rotation of the local coordinate system is proposed. According to this method, a multiscale macroscopic and mesoscopic finite element model of the No. 9 non-overflow dam section of Shimantan dam is constructed. The aggregate delivery simulation results show that the method presented in this paper can quickly generate two-dimensional (2D) random concrete aggregates, and the generation of three-dimensional (3D) aggregates can also be completed in a very short time, which can greatly improve the aggregate generation efficiency.
Some comments which greatly enhance the understanding of the paper and its value are presented below. Specific issues that require further consideration are:
- The title of the manuscript is matched to its content.
- In the Reviewer’s opinion, the current state of knowledge relating to the manuscript topic has been presented, but the author's contribution and novelty are not enough emphasized.
- Experimental program and results looks interesting and was clearly presented. Please provide a more detailed reasoning behind the behavior you have observed from the specimens.
- In the Reviewer’s opinion, the bibliography, comprising 59 references, is rather representative.
- An analysis of the manuscript content and the References shows that the manuscript under review constitutes a summary of the Author(s) achievements in the field. However, the introduction needs more attention.
- Please describe the process of each experiment. Also indicate the model of each tool that is used in the experiment.
- Conclusion needs to be more revised.
- In the Reviewer’s opinion the manuscript should be published in the journal after major revision.
Round 2
Reviewer 1 Report
Thank you for addressing all the comments sufficiently.
Reviewer 3 Report
Authors corrected articel follow to my suggestion. In my opionon should be published in the Journal.